# YOLO-MARL: You Only LLM Once for Multi-agent Reinforcement Learning

## Abstract

Advancements in deep multi-agent reinforcement learning (MARL) have positioned it as a promising approach for decision-making in cooperative games. However, it still remains challenging for MARL agents to learn cooperative strategies for some game environments. Recently, large language models (LLMs) have demonstrated emergent reasoning capabilities, making them promising candidates for enhancing coordination among the agents. However, due to the model size of LLMs, it can be expensive to frequently infer LLMs for actions that agents can take. In this work, we propose You Only LLM Once for MARL (YOLO-MARL), a novel framework that leverages the high-level task planning capabilities of LLMs to improve the policy learning process of multi-agents in cooperative games. Notably, for each game environment, YOLO-MARL only requires one time interaction with LLMs in the proposed strategy generation, state interpretation and planning function generation modules, before the MARL policy training process. This avoids the ongoing costs and computational time associated with frequent LLMs API calls during training. Moreover, the trained decentralized normal-sized neural network-based policies operate independently of the LLM. We evaluate our method across three different environments and demonstrate that YOLO-MARL outperforms traditional MARL algorithms.

## 1 Introduction

Multi-agent reinforcement learning (MARL) algorithms have proven to be a powerful framework for addressing complex decision-making problems in multi-agent systems. With the rising applications of multi-agent systems, such as mobile robots in warehouses and games requiring complex reasoning and strategy, it is increasingly crucial for individual agents to learn, cooperate, or compete in dynamic environments without a centralized decision-maker (Papoudakis & Schäfer, 2021). In cooperative Markov games, agents are trained to coordinate their actions to maximize the joint rewards. However, existing MARL algorithms face challenges in learning distributed policies for cooperative games. Moreover, they struggle with tasks characterized by sparse rewards, dynamic environment, and large action spaces, which can hinder efficient learning and agent collaboration.

LLMs have excelled as high-level semantic planners due to its in-context learning abilities and prior knowledge (Ahn et al., 2022). Zhang et al. (2023) and Kannan et al. (2024) directly use LLMs as embodied agents, which demonstrate LLMs' planning ability in multi-robot system. There are also works concentrating on utilizing the LLMs to guide the reinforcement learning (RL) training to reach better performances. ELLM (Du et al., 2023) leverage LLMs to suggest a goal to assist RL training whereas Kwon et al. (2023) focusing on the alignment between the action provided by LLM and the RL policy. While these approaches show exciting potential for integrating LLM within policy training, they have yet to extend their methods on multi-agent scenarios. More importantly, utilizing LLMs as agents or integrating them into the RL training loop presents certain challenges. Repeated interactions with LLMs in long-episode tasks or complex environments—especially when using advanced LLMs like Claude-3.5 or GPT-o1 can be time-consuming and costly; it becomes intractable for tasks requiring training over tens of millions of steps. Additionally, there is a risk of intermittent disconnections with the LLM, which could disrupt the training process and affect the system's stability.

Built on the identified insights and challenges, we introduce YOLO-MARL, as shown in Fig. 1, an innovative approach that leverages the planning capabilities of LLMs to enhance MARL pol-

---

*† These authors contributed equally to this work.

icy training. In particular, the major strength of our framework is that it requires only a one-time interaction with the LLM for each game environment. After the strategy generation, state interpretation and planning function generation modules, there is no need for further LLMs interaction during the MARL training process, which significantly reduces the communication and computational overhead of LLM inferences. Moreover, YOLO-MARL demonstrates its strong generalization capabilities and simplicity for application: with the proposed strategy generation and state interpretation modules, our approach is compatible with various MARL algorithms such as Yu et al. (2022), Rashid et al. (2018), Lowe et al. (2020), and requires only basic background understanding of a new game environment from the users. We also evaluate our framework in a sparser reward multi-agent environment: Level-Based Foraging environment (Papoudakis & Schäfer, 2021), and a highly strategic task environment: the StarCraft Multi-Agent Challenge environment (Samvelyan et al., 2019), together with the MPE environment (Lowe et al., 2020), and show that YOLO-MARL outperforms several MARL baselines. We also provide several ablation study results to demonstrate the function of each module in the proposed framework. To the best of our knowledge, YOLO-MARL is among one of the first trials that incorporates the high-level reasoning and planning abilities of LLMs with MARL, since very limited literature of LLM for MARL has been introduced so far (Sun et al., 2024).

In summary, our proposed method YOLO-MARL has the following advantages:

- This framework synergizes the planning capabilities of LLMs with MARL to enhance the policy learning performance in challenging cooperative game environments. In particular, our approach exploits the LLM's wide-ranging reasoning ability to generate high-level assignment planning functions to facilitate agents in coordination.
- YOLO-MARL requires minimal LLMs involvement, which significantly reduces computational overhead and mitigates communication connection instability concerns when invoking LLMs during the training process.
- Our approach leverages zero-shot prompting and can be easily adapted to various game environments, with only basic prior knowledge required from users.

An overview of YOLO-MARL is presented in Figure 1. All prompts, environments, and generated planning functions can be found in Appendix.

## 2 RELATED WORK

### 2.1 MULTI-AGENT REINFORCEMENT LEARNING

MARL has gained increasing attention due to its potential in solving complex, decentralized problems. Centralized training with decentralized execution has become a popular framework for overcoming the limitations of independent learning. Methods like QMIX (Rashid et al., 2018) and MADDPG (Lowe et al., 2020) use centralized critics or value functions during training to coordinate agents, while allowing them to execute independently during testing. In cooperative environments, algorithms like COMA (Foerster et al., 2017) and VDN (Sunehag et al., 2017) enable agents to share rewards and act in a coordinated fashion to maximize joint rewards.Wang et al. (2024) introduce a new approach using language constraint prediction to tackle the challenge of safe MARL in the context of natural language. However, the existing MARL algorithms may not perform well in sparse reward environments and still struggle in learning fully cooperative policy in some environments. So far, only very limited literature of using LLM for MARL has been proposed (Sun et al., 2024), and it remains unclear whether and how can LLM be leveraged for MARL-based decision-making.

### 2.2 LARGE LANGUAGE MODELS FOR SINGLE-AGENT RL AND DECISION-MAKING

Many existing works utilize LLMs as parts of RL training process. Du et al. (2023) enhance agents' exploration by computing the similarity between suggested goals from LLMs and agents' demonstrated behaviors. Carta et al. (2023) leveraging language-based goals from LLMs by generating actions conditioned on prompts during online RL. Kwon et al. (2023) provides scalar rewards based on suggestions from LLMs to guide RL training. However, most of these approaches haven't explored their works in the context of Markov games and require extensive interactions with LLMs during training.

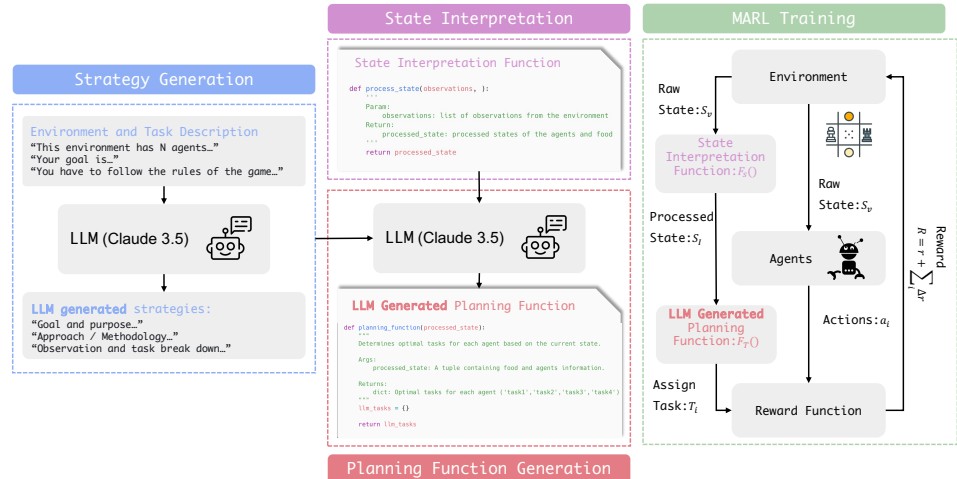

Figure 1: Depiction of our framework YOLO-MARL. (a). Strategy Generation: We pass basic environment and task description into the LLM to get generated strategies for this specific environment. (b). State Interpretation: We process the global states so that the format of global states will be more structured and organized for better comprehension by the LLM. (c). Planning Function Generation: We chain together the environment and task description, LLM generated strategies and state interpretation function. These prompts are then fed into the LLM to generate a Python planning function for this environment. (d). MARL Training: The state interpretation function and the generated planning function are integrated into the MARL training process. The LLM is no longer required for further interaction after the Planning Function Generation. The more detailed explanation of MARL Training part can be found in Algorithm 1

Gupta et al. (2022) utilize CLIP's visual embedding to an agent exploring of environment. Fan et al. (2022) studies a multi-task RL problem, where an agent is tasked with completing MINEDOJO tasks. Ahn et al. (2022) proposes SayCan which grounds LLMs via value functions of pretrained skills to execute abstract commands on robots. Liang et al. (2023) finds that code-writing LLMs can be re-purposed to write robot policy code. Huang et al. (2022) shows that by leveraging environment feedback, LLMs are able to form an inner monologue that allows them to more richly process and plan. Other research such as Ma et al. (2024) and Xie et al. (2023) use LLMs prior knowledge and code generation capability to generate reward functions, whereas we utilize code generation for planning functions. Lin et al. (2024) highlights the limitations of LLMs in handling complex low-level tasks. On the other hand, we harness the high-level reasoning capabilities of LLMs to enhance low-level action performance within RL model training.

### 2.3 LARGE LANGUAGE MODELS FOR MULTI-AGENT SYSTEMS

LLM-based Multi-Agent (LLM-MA) systems focus on diverse agent profiles, interactions, and collective decision-making. While this allows agents to collaborate on complex tasks, it also increases computational overhead due to the communication between LLMs (Guo et al., 2024), (Sun et al., 2024). Camel Li et al. (2024) and MetaGPT Hong et al. (2023) employ multiple LLM agents to accomplish tasks like brainstorming and software development. Nascimento et al. (2023) enhance communication and agent autonomy by integrating GPT-based technologies. In multi-robot contexts, Chen et al. (2023) compare task success rates and token efficiencies of four multi-agent communication frameworks. SMART-LLM (Kannan et al., 2023) decompose multi-robot task plans into subgoals for LLM to enable efficient execution, while Co-NavGPT (Yu et al., 2023) integrates LLMs as global planners for cooperative navigation. Focusing on multi-agent pathfinding (MAPF), Chen et al. (2024) studies the performance of solving MAPF with LLMs. Numerous studies have also focused on leveraging the decision-making capabilities of LLMs in complex computer game environments. (Hu et al., 2024). Agashe et al. (2023) introduced a benchmark for LLM-MA in coordination games. Gong et al. (2023) proposed an interactive framework and a novel environment that leverage LLMs as dispatchers for multi-agent system gaming. Wu et al. (2024) fine-tuned LLMs

based on gameplay outcomes, enabling them to adapt and improve their decision-making within the strategic game. Li et al. (2023) explore the use of LLMs in cooperative games within a text-based environment, and Ma et al. (2023) explores LLMs in the StarCraft II environment. In contrast, our method leverages the planning abilities of LLM to train better small-size neural network-based MARL policies instead of using LLMs directly as agents.

## 3 PROBLEM FORMULATION

**Markov game (MG)** is defined as a multi-agent decision-making problem when the interaction between multiple agents affect the state dynamics of the entire system and the reward of each agent under certain conditions (Littman, 1994). In this work, we consider a Markov game, or a stochastic game Owen (1982) defined as a tuple $G := (\mathcal{N}, S, A, \{r^i\}_{i \in \mathcal{N}}, p, \gamma)$, where $\mathcal{N}$ is a set of $N$ agents, $S = S^1 \times \cdots \times S^N$ is the joint state space, $A = A^1 \times \cdots \times A^N$ is the joint action space, with $(S^i, A^i)$ as the state space and action space of agent $i$, respectively, $\gamma \in [0, 1)$ is the discounting factor (Littman, 1994; Owen, 1982). The state transition $p : S \times A \rightarrow \Delta(S)$ is controlled by the current state and joint action, where $\Delta(S)$ represents the set of all probability distributions over the joint state space $S$. Each agent has a reward function, $r^i : S \times A \rightarrow \mathbb{R}$. At time $t$, agent $i$ chooses its action $a_t^i$ according to a policy $\pi^i : S \rightarrow \Delta(A^i)$. For each agent $i$, it attempts to maximize its expected sum of discounted rewards, i.e. its objective function $J^i(s, \pi) = \mathbb{E}\left[\sum_{t=1}^{\infty} \gamma^{t-1} r_t^i(s_t, a_t) | s_1 = s, a_t \sim \pi(\cdot | s_t)\right]$. In the literature, deep MARL algorithms (Lowe et al., 2020; Yu et al., 2022; Rashid et al., 2018) have been designed to train neural network-based policies $\pi_i(\theta_i)$. For a cooperative game, one shared reward function for all the agents is widely used during the training process, which is also considered in this work.

## 4 METHODOLOGY

In this section, we introduce our method, YOLO-MARL, which leverages LLMs to enhance MARL. Specifically, during training, we utilize the high-level task planning capabilities of LLMs to guide the MARL process. Our approach consists of four key components: Strategy Generation, State Interpretation, Planning Function Generation, and MARL training process with the LLM generated Planning Function incorporated throughout.

---

**Algorithm 1** YOLO-MARL Training Process

---

**Require:** Large Language Model $LLM$, State Interpretation function $F_S$, MARL actor $\mathcal{A}$, MARL algorithm $MARL_{alg}$, Initial Prompts $P_{init}$
1: **Hyperparameters:** reward signal $r'$, penalty signal $p'$
2: $P_{Strategy} \sim LLM(P_{init})$ // Strategy Generation
3: $P = P_{init} + P_{Strategy} + F_S$ // Chaining all the prompt for Planning Function Generation
4: $\mathcal{F}_{\mathcal{T}} \sim LLM(P)$ // Planning Function Generation: Sample functions code from the LLM
5: // MARL training with generated planning function
6: **for** each training step **do**
7:     $S_I \leftarrow F_S(S_v)$ // State Interpretation: Get processed global observation $S_I$ from $F_S$
8:     $\mathcal{T}_1, \mathcal{T}_2, \ldots \leftarrow \mathcal{F}_{\mathcal{T}}(S_I)$ // Assign tasks $\mathcal{T}$ to each agent
9:     $a_1, a_2, \ldots \leftarrow \mathcal{A}(S_v)$ // Output actions from the actor
10:     **for** each agent $i$ **do**
11:         **if** $a_i \in \mathcal{T}_i$ **then**
12:             $\Delta r_i \leftarrow r'$
13:         **else**
14:             $\Delta r_i \leftarrow p'$
15:         **end if**
16:     **end for**
17:     $R \leftarrow r + \sum_i \Delta r_i$ // Compute final reward for criticz: More details are in equation 1, 2
18:     $\pi(\theta) = MARL_{alg}(R)$ // Use $R$ as the final reward for MARL training
19: **end for**
20: **return** Trained MARL policy

---

### 4.1 STRATEGY GENERATION

To create a generalizable framework applicable to various environments—especially when users may have limited prior knowledge—we incorporate a Strategy Generation Module into our methodology. This module enables the LLM to autonomously generate strategies for different environments without requiring extensive human input or expertise.

As shown inside the blue box of Figure 1(a), the LLM is provided with the basic information about the environment, including task descriptions, relevant rules, and constraints of how to interact with the environment. Additionally, we supply a general guideline within the prompt to assist the LLM in generating effective strategies. Gathering all the information, the LLM will output detailed strategies to accomplish the tasks or achieve the goal, following the specified format.

By aggregating all this information, the LLM outputs detailed strategies to accomplish the tasks or achieve the goals, following a specified format. The Strategy Generation is crucial for several reasons:

- Reducing User Burden: It alleviates the need for users to comprehensively understand new environments, saving time and effort.
- Enhancing Generalization: It enables the framework to adapt to different environments with minimal prompt modifications.
- Facilitating Planning Function Generation: The strategies serve as vital components in the prompts used for the Planning Function Generation Module. The results of using YOLO-MARL but without Strategy Generation Module are shown in ablation study 6.1 .

The LLM-generated strategies are incorporated into the prompt alongside other necessary information to facilitate the subsequent planning function generation. Further details about the strategy prompts and their formats can be found in Appendix C.1.

## 4.2 STATE INTERPRETATION

In many simulation environments, observations or states are typically provided as vectors, with each component constructed using various encoding methods. While the vector form of observation is easy to handle when training deep reinforcement learning models, it is difficult for LLMs to directly parse their semantic meaning due to the lack of explicit context for each component.

We propose the State Interpretation Module to assist the LLM in interpreting the environment state. By providing a semantically meaningful representation of the state, the LLM can successfully generate executable planning functions for training. Formally, given the current environment state in vector form $S_v$, we define an interpretation function $F_S$ such that $F_S(S_v) \to S_I$, where $S_I$ provides more explicit and meaningful information about each state component.

Recent works like Ma et al. (2024) and Xie et al. (2023) have demonstrated the success of enhancing LLMs performance by providing relevant environment code. In the same manner, we include the interpretation function $F_S$ in the prompting pipeline, formatted as Pythonic environment code as shown in the purple box in Figure 1(b). The State Interpretation Module significantly reduces the risk of the LLM generating erroneous functions with outputs incompatible with the training procedures. An ablation study on the effectiveness of this module can be found in Sec 6.2, while more details about the interpretation function are provided in Appendix C.2.

## 4.3 PLANNING FUNCTION GENERATION

A crucial component of our method is leveraging the LLM to perform high-level planning instead of handling low-level actions. We combine all the prompts from the previous modules and input them into the LLM. The LLM then generates a reasonable and executable planning function that can be directly utilized in the subsequent training process.

To be more concise, given any processed state $S_I$, we define an assignment planning function as $\mathcal{F}_{\mathcal{T}}(S_I) \to \mathcal{T}_i \in \mathcal{T}$, where $\mathcal{T} = \{\mathcal{T}_1, ..., \mathcal{T}_n\}$ is a set of target assignments that each agent can take. We define the assignment set $\mathcal{T}$ over the action space such that an action can belong to multiple assignments and vice versa. For example, if the assignment space is defined as $\mathcal{T} = \{Landmark\_0, Landmark\_1\}$, and landmark 0 and landmark 1 are located at the top right and top left positions relative to the agent respectively, then taking the action "UP" can be associated with both assignments. Conversely, we can have multiple actions correspond to an assignment. For instance, moving towards "Landmark 0" may involve actions like "UP" and "RIGHT".

The planning function generation will only be required once for each new environment you try to use. After you interact with the LLM to get generated planning function, you can directly use it

in the later training process with different MARL algorithms. This is referred to the red module in Fig. 1(c) and more information of generated function refer to Appendix D.

## 4.4 MARL TRAINING WITH PLANNING FUNCTION INCORPORATION

To incorporate the planning function into MARL training, we add an extra reward term to the original reward provided by environments. Specifically, we define the final reward $R$ used by the critic as:

$$R = r + \sum_i \Delta r_i. \tag{1}$$

Here, $r$ is the original reward from the environment. For each agent $i$, $\Delta r_i$ is an additional reward or penalty that determined based on whether the action taken by the agent aligns with the task assigned by the planning function. Specifically:

$$\Delta r_i = \begin{cases} r', \text{if the agent i action aligns with the assigned task,} \\ p', \text{if the agent i action doesn't align with the assigned task.} \end{cases} \tag{2}$$

Notably, we don't need to interact with the LLM during the entire training process, nor do we need to call the planning function after the policy has been trained. The training process $MARL_{alg}(R)$ takes $R$ as the reward function, uses the same state and action space. We follow the standard MARL algorithms and evaluation metrics within the literature, such as Yu et al. (2022), Rashid et al. (2018), and Lowe et al. (2020). Our method, as shown in the greed box in Fig.1(d), is highly efficient compared to approaches that interact with LLMs throughout the whole training process or directly use LLMs as agents. In practice, using the LLM's API to generate the planning function incurs minimal cost—less than a dollar per environment—even when using the most advanced LLM APIs.

## 5 EXPERIMENTS

In this section, we evaluate our method across three different environments: MPE, LBF, and SMAC. We use `claude-3-5-sonnet-20240620` for the experiments.[*]

### 5.1 SETUP

**Baselines.** In our experiments, we compare the MARL algorithm MADDPG (Lowe et al., 2020), MAPPO (Yu et al., 2022) and QMIX (Rashid et al., 2018) and set default hyper-parameters according to the well-tuned performance of human-written reward, and fix that in all experiments on this task to do MARL training. Experiment hyper parameters are listed in Appendix.

**Metrics.** To assess the performance of our method, we use win rate as the evaluation metric on the SMAC environment, and the mean return in evaluation for all other environments. During evaluation, we rely solely on the default return values provided by the environments for both the baseline and our method, ensuring a fair comparison.

### 5.2 RESULTS

**Level-Based Foraging.** Level-Based Foraging (LBF) (Papoudakis & Schäfer, 2021) is a challenging sparse reward environment designed for MARL training. In this environment, agents must learn to navigate a path and successfully collect food, with rewards only being given upon task completion. To evaluate our framework in a cooperative setting, we selected the 2-player, 2-food fully cooperative scenario. In this setting, all agents must work together and coordinate their actions to collect the food simultaneously. The environment offers an action space consisting of [NONE, NORTH, SOUTH, WEST, EAST, LOAD], and we define the task set as [NONE, Food i, ..., LOAD]. Using the relative positions of agents and food items, we map assigned tasks to the corresponding actions in the action space and calculate the reward based on this alignment. We evaluated our framework over

---

[*]We mainly use the Claude 3.5 Sonnet model for the LLM in our work: `https://www.anthropic.com/news/claude-3-5-sonnet`

Table 1: Comparison between YOLO-MARL and MARL in the LBF environment across three seeds. The highest evaluation return means during training are highlighted in bold. The corresponding results can be found in Figure 2. The M means one million training steps. We run all the experiments on the same machine.

| | Mean Return after 0.2M / 0.4M / 1.5M / 2M Steps | | |
| --- | --- | --- | --- |
| | QMIX | MADDPG | MAPPO |
| MARL | 0.00/ 0.01/ 0.25/ 0.38 | 0.09/ 0.33/ 0.26/ 0.32 | 0.31/ 0.72/ 0.99/ 0.99 |
| YOLO-MARL | **0.01/ 0.02 / 0.60/ 0.78** | **0.13/ 0.38/ 0.39/ 0.44** | **0.93/ 0.98/ 0.99/ 0.99** |

3 different seeds, with the results shown in Figure 2 and Table 1. LLM assist the MARL algorithm by providing reward signals, our framework significantly outperformed the baseline, achieving a maximum improvement of **105 %** in mean return and a **2x faster** convergence rate among all tested MARL algorithms. According to the results, our framework is effective across all the baseline algorithms, with particularly large improvements observed in QMIX and MADDPG, and a faster convergence rate for MAPPO. To assess the variability in the quality of our generated functions, we present the results of three different generated functions in Figure 8 and Table 3 in Appendix B.1. The results demonstrate that our framework consistently generates high-quality functions, with each achieving similar improvements across all baseline algorithms.

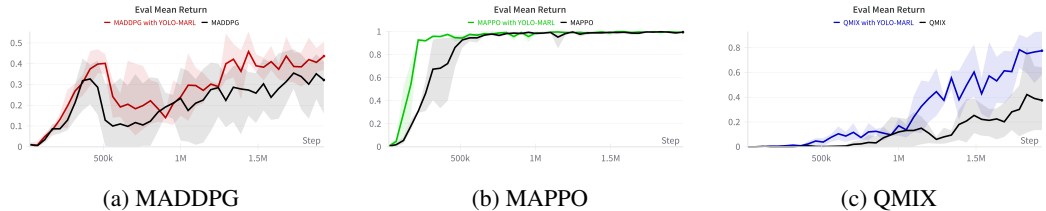

(a) MADDPG        (b) MAPPO        (c) QMIX

Figure 2: **Results for LBF environment across 3 seeds:** The solid lines indicate the mean performance, and the shaded areas represent the range (minimum to maximum) across 3 different seeds.

**Multi-Agent Particle Environment.** We evaluate our framework in Multi-Agent Particle Environment (MPE) (Lowe et al., 2020) simple spread environment which is a fully cooperative game. This environment has N agents, N landmarks. At a high level, agents must learn to cover all the landmarks while avoiding collisions. It's action space is consist of [no_action, move_left, move_right, move_down, move_up]. We define the assignment for each agent to take to be [Landmark_i,...,No action]. During training, based on the global observation, we obtain the relative position of each agent with respect to the landmarks. Similar to LBF, we map each assignment of agent back to the corresponding action space and then reward the action of policy in action space level. We evaluate our approach on 3-agent and 4-agent scenarios using QMIX and MADDPG as baselines. As shown in Figure 3, our framework(colored line) outperform the baseline(black line) algorithm in mean returns by **7.66%** and **8.8%** for 3-agent scenario, and **2.4%** and **18.09%** for 4-agent scenario with QMIX and MADDPG respectively. These improvements demonstrate the effectiveness of our framework in enhancing coordination among agents to cover up all the landmarks.

**StarCraft Multi-Agent Challenge environment.** The StarCraft Multi-Agent Challenge (SMAC) (Samvelyan et al., 2019) simulates battle scenarios where a team of controlled agents must destroy an enemy team using fixed policies within a limited number of steps. We tested our method on three different maps: 3M, 2s vs 1sc, and 2c vs 64zg. The action space in the environment consists of [none, stop, move north, move south, move west, move east, attack enemy 1,...attack enemy n], where n is the total number of enemies on the map. This action space becomes increasingly complex depending on the number of enemies the agent has to engage, particularly in the 2c vs 64zg map, which contains 64 enemies and offers 70 possible actions.

In our experiments, we define the assignment space simply as [Move, Attack, Stop, None (for dead agents)]. We tested the performance of MAPPO, and the results for SMAC are shown in Figure 5. As indicated by the figure, even though we provide simple assignments that may be far from optimal

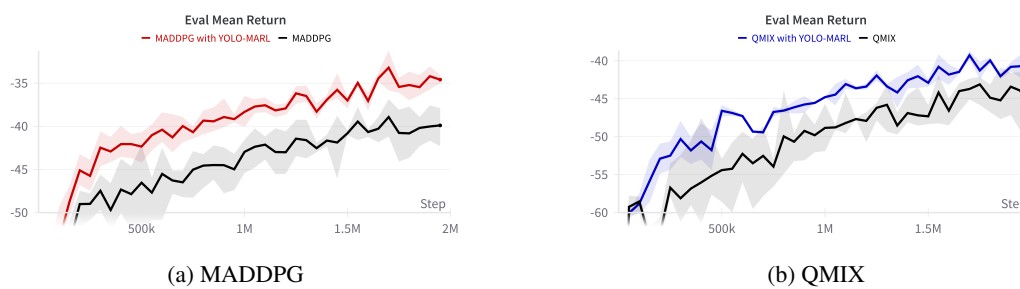

(a) MADDPG                                (b) QMIX

Figure 3: MPE simple spread scenario 3 agents results. The solid lines indicate the mean performance, and the shaded areas represent the range (minimum to maximum) across 3 different generated planning function.

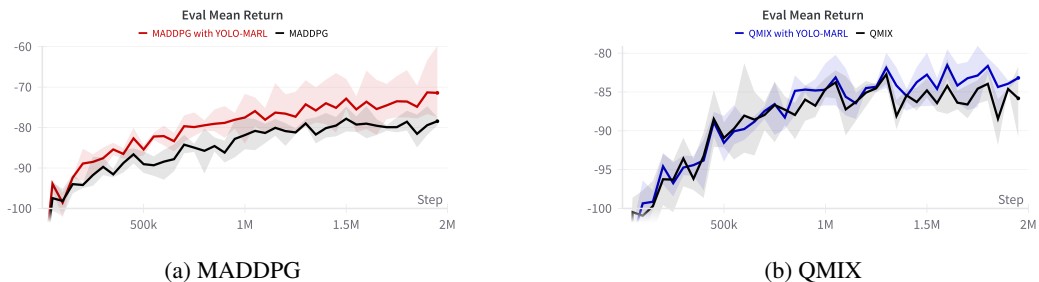

(a) MADDPG                                (b) QMIX

Figure 4: MPE simple spread scenario 4 agents results. The solid lines indicate the mean performance, and the shaded areas represent the range (minimum to maximum) across 3 different generated planning function.

instructions, our framework still achieves comparable results on certain maps. This demonstrates that our framework remains competitive, even in environments requiring strategic movements. We also explore the sparse reward case for this environment where the win rate of baseline algorithms is always closed to 0 while we generate a planning reward function pairs that outperform baseline. We suggest this pair generation as a potential future work and leave this discussion to the Sec 7.

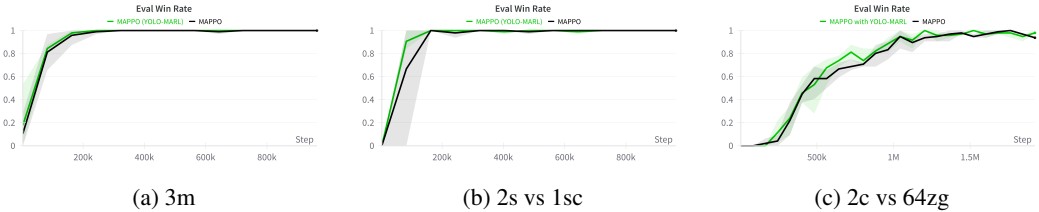

(a) 3m                       (b) 2s vs 1sc                     (c) 2c vs 64zg

Figure 5: Results for 3 maps on SMAC environment: Average win rate comparison with our method for MAPPO baseline on 3 maps: 3m, 2s vs 1sc and 2c vs 64zg across 3 different seeds and the solid lines indicate the mean performance.

## 6 ABLATION STUDY

In this section, we conduct the ablation studies mainly in LBF 2 players 2 food fully cooperative environment since rewards in LBF are sparser compared to MPE and SMAC (Papoudakis & Schäfer, 2021). We refer to 5.2 for more information about the environment. Due to page limitation, we also leave some discussions and figures in Appendix B.

## 6.1 COMPARISON BETWEEN YOLO-MARL WITH AND WITHOUT STRATEGY GENERATION

In this section, we examine the impact of the Strategy Generation Module on the performance of the YOLO-MARL framework. Specifically, we compare the standard YOLO-MARL with a variant that excludes the Strategy Generation Module to assess its significance.

According to our tests, the Strategy Generation Module plays an important role in the YOLO-MARL method. As shown in Figure 6 , without the LLM generated strategy, we obtain a worse-performing planning function. Interestingly, the mean returns of evaluations for the functions without the LLM generated strategy are not always close to zero, indicating that the generated planning functions are not entirely incorrect. Based on this, we could confirm that the Strategy Generation Module would help Planning Function Generation Module provides better solutions to this game. Moreover, giving the strategy also helps stabilize the quality of the generated code. We observe a higher risk of obtaining erroneous functions without supplying the strategy.

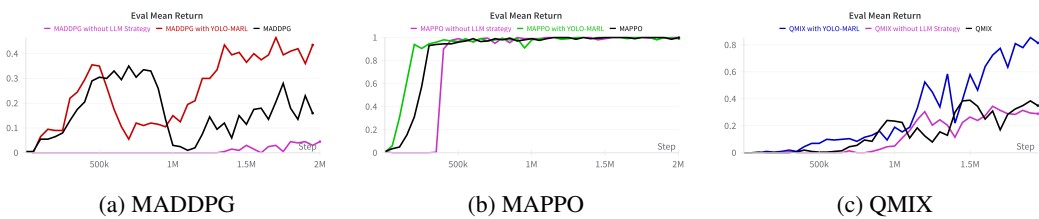

|     (a) MADDPG     |     (b) MAPPO     |     (c) QMIX     |

Figure 6: Comparison between YOLO-MARL with and without using LLM generated strategies in LBF

## 6.2 COMPARISON BETWEEN YOLO-MARL WITH AND WITHOUT STATE INTERPRETATION

To demonstrate how the State Interpretation Module enhances our framework, we present two failure case snippets:

- Without the Interpretation Function: The interpretation function is omitted entirely from the prompting pipeline.
- Providing Raw Environment Code Directly: Raw environment source code is fed directly to the LLM.

As shown in Figure 10, the LLM is unable to infer the type of state and attempts to fetch environment information via a non-existent key if no preprocessing code provided. And if environment code is provided without dimensional context for each component, the LLM is likely to make random guesses. In both scenarios, the absence of explicit state interpretation hinders the LLM's ability to generate accurate and executable planning functions. These failures underscore the importance of the State Interpretation Module in bridging the gap between vectorized observations and the LLM's requirement for semantically meaningful input.

By incorporating the State Interpretation Module, we enable the LLM to understand the environment's state representation effectively. This results in the generation of reliable planning functions that significantly enhance the performance of our YOLO-MARL framework.

## 6.3 COMPARISON BETWEEN YOLO-MARL AND REWARD GENERATION

In this section, we compare our YOLO-MARL method with approaches that utilize the LLM for reward generation without reward function template. We explore two scenarios: reward generation without feedback and reward generation with feedback. For the reward generation without feedback, the reward function is generated at the same stage as the planning function for fair comparison. This means that we generate the reward function before all the training process for each new environment. For the reward generation with feedback, we first generate a reward function just like the reward generation without feedback. And then, iteratively, we will run a whole training process on this

environment and pass the feedback of this training performance to the LLM, combined with previous prompts and ask the LLM to refine the previous generated reward function.

Our experiments show that relying solely on the LLM-generated reward function leads to poor performance. As shown in Figure 7, the mean return for the LLM-generated reward function pair consistently falls below the performance of all three MARL algorithms. This indicates that agents are not learning effectively under the LLM-generated reward function. However, we do observe a slight positive return. This suggest the potential of using this framework for reward shaping tasks, particularly in situations where standard MARL algorithms struggle to learn in sparse reward scenarios. To investigate whether iterative refinement could improve the LLM generated reward function,

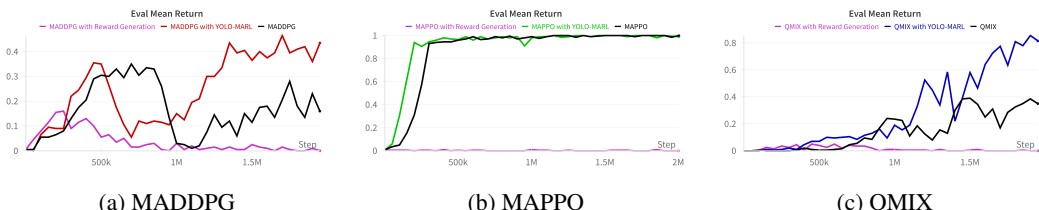

| (a) MADDPG | (b) MAPPO | (c) QMIX |

Figure 7: Comparison between YOLO-MARL and reward generation without feedback in LBF

we supply the LLM with the generated reward function from the prior iteration and feedback on its performance. Despite this iterative process, the LLM still fails to output a suitable reward function for the LBF environment. The mean return of evaluations remains close to zero, as shown in figure 9. The generated reward functions for each iteration are provided in Appendix E.

## 7 LIMITATION AND FUTURE WORK

We acknowledge that the performance of YOLO-MARL may be highly correlated with the LLM's ability and we haven't tested YOLO-MARL with other LLMs like GPT-o1 due to the tier5 user requirement, and there might be a gap of YOLO-MARL's performance between the Claude-3.5 and GPT-o1.

For future work, we are enthusiastic about the potential for LLMs to further enhance MARL, particularly as their planning capabilities improve. Specifically, we envision combining reward generation with planning functions to boost the performance of existing MARL algorithms in fully sparse environments. In this approach, we prompt the LLM to generate both a planning function and a reward function that replaces the environment-provided reward, following the pipeline described in Section 4. The function-pair method may require further refinement, and we will explore it as a future direction. A preliminary test of this framework is provided in Appendix B.4.

## 8 CONCLUSION

We propose YOLO-MARL, a novel framework that leverages the high-level planning capabilities of LLMs to enhance MARL policy training for cooperative games. By requiring only a one-time interaction with the LLM for each environment, YOLO-MARL significantly reduces computational overhead and mitigates instability issues associated with frequent LLM interactions during training. This approach not only outperforms traditional MARL algorithms but also operates independently of the LLM during execution, demonstrating strong generalization capabilities across various environments.

We evaluate YOLO-MARL across three different environments: the MPE environment, the LBF environment, and the SMAC environment. Our experiments showed that YOLO-MARL outperforms or achieve competitive results compared to baseline MARL methods. The integration of LLM-generated high-level assignment planning functions facilitated improved policy learning in challenging cooperative tasks, even in environments characterized by sparser rewards and large action spaces. Finally, we mention a possible way to incorporate reward generation to our framework and we will step further.

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

# APPENDIX

## A    HYPERPARAMETER DETAILS

The detail hyper-parameter for the baseline algorithm can be found in Yu et al. (2022) and Papoudakis & Schäfer (2021). We provide the full hyper-parameters for the reward and penalty value given to the RL training throughout the experiments in 2.

Table 2: Hyperparameter

| | | LBF | | |
|---|---|---|---|---|
| | | QMIX | MADDPG | MAPPO |
| | r' | $0.02 \pm 0.01$ | $0.002 \pm 0.001$ | $0.005 \pm 0.004$ |
| | p' | $0.02 \pm 0.01$ | $0.002 \pm 0.001$ | $0.005 \pm 0.004$ |

| | MPE(3agents/4agents) | | SMAC (3m/2s_vs_1sc/2c_vs_64zg) |
|---|---|---|---|
| | MADDPG | QMIX | MAPPO |
| r' | $0.2/0.3 \pm 0.1$ | $0.2 \pm 0.1/0.2 \pm 0.1$ | $0.001 \sim 0.01 / 0.02 / 0.003 \pm 0.002$ |
| p' | $0.1/0.2 \pm 0.1$ | $0.2 \pm 0.1/0.2 \pm 0.1$ | $0.001 \sim 0.01 / 0.02 / 0.003 \pm 0.002$ |

## B    ADDITIONAL RESULT

Given the page constraints, we present some additional experiments and ablation study results and figures in this section.

### B.1    COMPARISON FOR DIFFERENT GENERATED FUNCIONS

Considering the variation on the output of LLMs, we evaluate the quality of generated functions and compare the results on 3 baseline methods and those using our framework. We conduct the experiments in LBF environment introduce in Sec 5.2

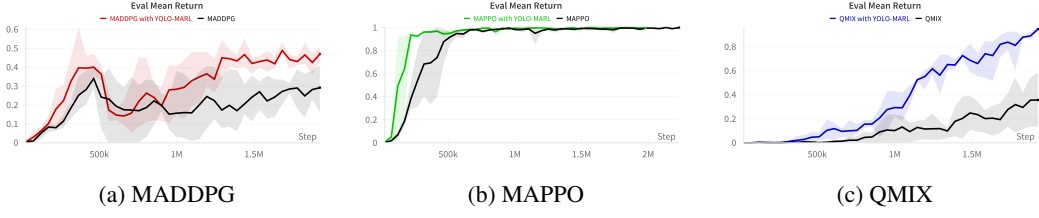

| (a) MADDPG | (b) MAPPO | (c) QMIX |
|---|---|---|

Figure 8: **Results for LBF environment across 3 seeds:** The solid lines indicate the mean performance, and the shaded areas represent the range (minimum to maximum) across 3 different seeds.

### B.2    ADDITIONAL RESULTS FOR REWARD FEEDBACK

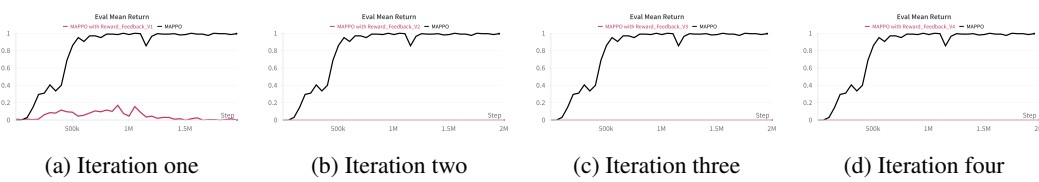

| (a) Iteration one | (b) Iteration two | (c) Iteration three | (d) Iteration four |
|---|---|---|---|

Figure 9: Results of only reward generation with feedback in the LBF environment. The total number of iteration is 4 and the MARL algorithm we used here is MAPPO.

Table 3: Comparison between YOLO-MARL and MARL in the LBF environment across three different generated planning functions. The highest evaluation return means during training are highlighted in bold. The corresponding results can be found in figure 8. The M means one million training steps. We use two different machines to generate planning functions and run MARL and YOLO-MARL on the same machines where the planning functions are generated.

| | Mean Return after 0.2M / 0.4M / 1.5M / 2M Steps | | |
| --- | --- | --- | --- |
| | QMIX | MADDPG | MAPPO |
| MARL | 0.00/ 0.01/ 0.25/ 0.36 | 0.08/ 0.28/ 0.24/ 0.29 | 0.38/ 0.74/ 0.99/ 0.99 |
| YOLO-MARL | **0.00/ 0.03/ 0.69/ 0.95** | **0.18/ 0.40/ 0.42/ 0.47** | **0.94/ 0.97/ 0.99/ 0.99** |

### B.3 ADDITIONAL RESULTS FOR STATE INTERPRETATION ABLATION STUDY

```
1   def planning_function(state):
2       ...
3       for agent_id, agent_info in own_info.items():
4           if agent_info['health'] <= 0:
5               llm_tasks[agent_id] = 'None'
6               continue
7           closest_enemy = min(enemy_info.values(), key=lambda x: ((x['x'] - agent_info['x'])**2 + ...
8       ...
9       return llm_tasks
```

(a) Failure Case: Without providing interpretation code

```
1   def planning_function(state):
2       ...
3       # Extract relevant information
4       move_feats = agent_state[:8]  # Assuming 8 movement features
5       enemy_feats = agent_state[8:8+5*n_enemies].reshape(n_enemies, 5)
6       ally_feats = agent_state[8+5*n_enemies:8+5*n_enemies+5].reshape(1, 5)
7       own_feats = agent_state[-5:]
8       ...
9       return llm_tasks
```

(b) Failure Case: Feeding environment code directly

Figure 10: Failure cases for YOLO-MARL without State Interpretation Module

### B.4 ADDITIONAL RESULT ON FUTURE WORK

We tested this new approach that utilizing YOLO-MARL to generate planning and reward function pair in the SMAC environment with a fully sparse reward setting. The baselines tested on the three SMAC maps performed poorly, with evaluation win rates consistently near zero. However, as demonstrated in Figure 11, incorporating the planning function into reward generation significantly improved performance.

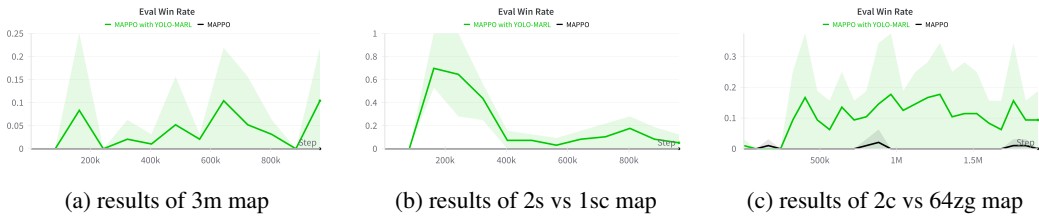

(a) results of 3m map     (b) results of 2s vs 1sc map     (c) results of 2c vs 64zg map

Figure 11: YOLO-MARL reward generation paired with planning function in SMAC under sparse reward setting

## C PROMPT DETAIL

In this section, we provide a comprehensive overview of the prompts used throughout the research/application to facilitate various tasks. The prompts play a critical role in guiding the behavior of language models or agents by providing them with specific instructions and constraints. This section details the exact wording, format, and context of the prompts that were used to achieve the results described in the main body of the paper.

### C.1 STRATEGY

The prompt for the strategy generation is consisted of **Environment Description**, **Assignment Class** and **Instruction**. **Environment Description** is about the environment information, we only provide some necessary description on what is this environment look like, what's the goal for the tasks. We also add the rules for some additional information or constraint for the game that should be followed and they can be found on the official website. **Assignment Class** can be viewed as splitting up the action space or sub goals that LLM could assigned to agent during the task, the formal definition can be found on Section 4.3. The **Instruction** is basically to tell what llm should output for the strategy. Below we provide the sample prompt for the most simple scenario in each environment, but prompt for rest of all scenarios is in the similar format following the these prompts.

**Level-Based Foraging**

---

**Environment Description:**
This Level-Based Foraging (LBF) multi-agent reinforcement learning environment has 2 agents and 2 food items. Your goal is to make the agents collaborate and pick up all the food present in the environment.

**Game Rules:**

1. The Pickup action is successful if all the agents pick up the same target together.
2. The Pickup action is only successful if the sum of the levels of the agents is equal to or higher than the level of the food.
3. The Pickup action is only allowed if the agents are within a distance of 1 relative to the food.
4. Success Condition: All food must be picked up before {time_steps} steps.

**Tasks Assignment:** Available tasks for each agent:

1. Target food 0
2. Target food 1
3. Pickup

**Instruction Format:** Here is a general guideline for generating strategies:

1. **Goal or Purpose**: Clearly state the overall objective of the task.
2. **Problem or Need**: Consider different scenarios and identify the key problem or need that the task plan addresses.
3. **Approach / Methodology**: Describe the overall approach or methodology step-by-step that will be followed.
4. **Scenario Analysis**: Consider different scenarios that agents could encounter during task execution and how they will coordinate to adapt.
5. **Task Breakdown**: Break down tasks, detailing the roles and responsibilities of each agent and how they will coordinate to achieve the overall objective.

---

**Multi-Agent Particle Environment**

**Environment Description:**
This Multi-Agent Particle Environment (MPE) multi-agent reinforcement learning environment has 3 agents and 3 landmarks. Your goal is to make agents collaborate and cover all the landmarks. **Game Rules:**

1. Agents must cover all landmarks by minimizing the distances between each landmark, with each agent going to a unique landmark.

2. Agents cannot collide with another agent. The collision threshold is 0.3.

**Tasks Assignment:** Available tasks for each agent:

1. Landmark 0

2. Landmark 1

3. Landmark 2

4. No op

**Instruction Format:** Here is a general guideline for generating strategies:

1. **Goal or Purpose**: Clearly state the overall objective of the task.

2. **Problem or Need**: Consider different scenarios and identify the key problem or need that the task plan addresses.

3. **Approach / Methodology**: Describe the overall approach or methodology step-by-step that will be followed.

4. **Scenario Analysis**: Consider different scenarios that agents could encounter during task execution and how they will coordinate to adapt.

5. **Task Breakdown**: Break down tasks, detailing the roles and responsibilities of each agent and how they will coordinate to achieve the overall objective.

**StarCraft Multi-Agent Challenge Environment**

**Environment Description:**
This SMAC 3m map has 3 Terran Marines agents and 3 Terran Marines enemies. The Agent unit is Marines, and its feature is that Marines are ranged units that can attack ground and air units. They are the basic combat unit for Terran and are versatile in combat. Your task is to utilize the unit information to win the battle scenario within 60 steps.

**Game Rules:**

1. Shooting range is 6 and sight range is 9 for both agent and enemy.

2. Success condition: Eliminate all enemy units before the episode ends.

3. Failure condition: If agents aren't aggressive enough to kill all the enemies to win within 60 steps, or if all agents die.

**Tasks Assignment:** Available tasks for each agent:

1. Move

2. Attack

3. Stop

4. None (only for dead agents)

**Instruction Format:** Here is a general guideline for generating strategies:

1. **Goal or Purpose**: Clearly state the overall objective of the task.

2. **Problem or Need**: Consider different scenarios and identify the key problem or need that the task plan addresses.

3. **Approach / Methodology**: Describe the overall approach or methodology step-by-step that will be followed.

4. **Scenario Analysis**: Consider different scenarios that agents could encounter during task execution and how they will coordinate to adapt.

5. **Task Breakdown**: Break down tasks, detailing the roles and responsibilities of each agent and how they will coordinate to achieve the overall objective.

## C.2 INTERPRETATION FUNCTION

Here we list the Interpretation Function for each scenerios that process the raw vector observation.

**LBF 2 player 2 food scenerio**

```python
def process_state(observations, p=2, f=2):
    '''
    Param:
        observation:
                            array of array (p, n): dict('agent_0', 'agent_1',
    ..., 'agent_p')
                            List:
                            Agent : (n, ) list of observation components
        p: int, number of agents
        f: int, number of foods in the environment
    Return:
        obs: tuples (food_info, agents_info):
            food_info: dictionary that contains information about food in
     the environment
                            key: food_id ('food_0', 'food_1', ...)
                            value: tuples (food_pos, food_level) or None if
    the food is already been picked up
            agents_info: dictionary that contains information about
    agents in the environment
                            key: agent_id ('agent_0', 'agent_1', ...)
                            value: tuples (agent_pos, agent_level)
    '''
    food_info = {}
    agents_info = {}
    obs = observations[0]
    offset = 0
    for food_idx in range(f):
        food_obs = obs[offset:offset+3]
        offset += 3
        curr_food_pos = food_obs[:2]
        curr_food_level = food_obs[2]
        food_id = f'food_{food_idx}'
        # If food level is 0, then the food is already been pickup and
    not present in the environment
        if curr_food_level == 0 and curr_food_pos[0] < 0:
            food_info[food_id] = None
        # The food is present in the environment
        else:
            food_info[food_id] = (curr_food_pos, curr_food_level)

    for agent_idx in range(p):
        agent_obs = obs[offset:offset+3]
        offset += 3
        curr_agent_pos = agent_obs[:2]
        curr_agent_level = agent_obs[2]
        agent_id = f'agent_{agent_idx}'
```

```
42        agents_info[agent_id] = (curr_agent_pos, curr_agent_level)
43
44    return food_info, agents_info
```

**MPE 3 agents scenerio**

```
1  def process_state(observations, N=3):
2      '''
3      Param:
4          observations:
5              List of NumPy arrays, one per agent.
6              Each array represents the observation for an agent:
7              [self_vel (2,), self_pos (2,), landmark_rel_positions (N*2,),
    other_agent_rel_positions ((N-1)*2,), communication]
8
9      Return:
10         obs:
11             Dictionary with agent IDs as keys ('agent_0', 'agent_1', ...)
    .
12             Each value is a list containing:
13                 - Landmark relative positions: N arrays of shape (2,)
14                 - Other agents' relative positions: (N-1) arrays of shape
    (2,)
15     '''
16     obs = {}
17     num_agents = len(observations)
18
19     for idx, agent_obs in enumerate(observations):
20         agent_id = f'agent_{idx}'
21         obs[agent_id] = []
22
23         # Extract landmark relative positions
24         for i in range(N):
25             start = 4 + 2 * i
26             end = start + 2
27             land_2_a = agent_obs[start:end]
28             obs[agent_id].append(land_2_a)
29
30         # Extract other agents' relative positions
31         for i in range(num_agents - 1):
32             start = 4 + 2 * N + 2 * i
33             end = start + 2
34             other_agent_2_a = agent_obs[start:end]
35             obs[agent_id].append(other_agent_2_a)
36
37     return obs
```

**MPE 4 agents scenerio**

```
1  def process_state(observations, N=4):
2      '''
3      Param:
4          observations:
5              List of NumPy arrays, one per agent.
6              Each array represents the observation for an agent:
7              [self_vel (2,), self_pos (2,), landmark_rel_positions (N*2,),
    other_agent_rel_positions ((N-1)*2,), communication]
8
9      Return:
10         obs:
11             Dictionary with agent IDs as keys ('agent_0', 'agent_1', ...)
    .
12             Each value is a list containing:
```

```
13                 - Landmark relative positions: N arrays of shape (2,)
14                 - Other agents' relative positions: (N-1) arrays of shape
    (2,)
15     '''
16     obs = {}
17     num_agents = len(observations)
18
19     for idx, agent_obs in enumerate(observations):
20         agent_id = f'agent_{idx}'
21         obs[agent_id] = []
22
23         # Extract landmark relative positions
24         for i in range(N):
25             start = 4 + 2 * i
26             end = start + 2
27             land_2_a = agent_obs[start:end]
28             obs[agent_id].append(land_2_a)
29
30         # Extract other agents' relative positions
31         for i in range(num_agents - 1):
32             start = 4 + 2 * N + 2 * i
33             end = start + 2
34             other_agent_2_a = agent_obs[start:end]
35             obs[agent_id].append(other_agent_2_a)
36
37     return obs
```

**SMAC 3m map**

```
1  def process_global_state(global_state, n=3, m=3):
2      '''
3      Param:
4          observation:
5                          Dict of list of (n, ): dict('agent_0', 'agent_1',
      ..., 'agent_N')
6                          List:
7                          Agent : (m, ) list of observation components
8          n: int, number of agents
9          m: int, number of enemies
10     Return:
11         obs (tuples of dict): Tuples of dict of (n, ): Tuple of each
      observation components processed from each agent's perspective by
      function "process_observation":
12             available_move_actions (dict of list): Dict of list of (4, ):
       dict('agent_0', 'agent_1', ..., 'agent_N') List of available moves
      for each agent. This might be empty if the agent is dead or no
      available move direction.
13                 ->available_move_actions[agent_id]: the available
      list looks like list of string ["North", "South", "East", and "West"]
       directions
14             enemy_info (dict of dict of tuple): Dict of dict of tuple of
      (n, ): dict('agent_0', 'agent_1', ..., 'agent_N') Tuple of m enemies
      information(enemy_0 to enemy_m) for each agent.
15                 ->enemy_info[agent_id][enemy_id]: each tuple contains
       information of (is current enemy available to attack, distant to
      current enemy, x direction position to current enemy, y direction
      position to current enemy, is current enemy visible, enemy health,
      enemy's x pos to center, enemy's y pos to center)
16             ally_info (dict of dict of tuple): Dict of dict of tuple of (
      n, ): dict('agent_0', 'agent_1', ..., 'agent_N') Tuple of n-1 ally
      information(exclude self) for each agent.
17                 ->ally_info[agent_id][al_id]: each tuple contains
      information of (is current ally visible, distant to current ally, x
      direction position to current ally, y direction position to current
```

```
        ally, ally's attack cooldown condition, ally's health, ally's x pos
        to center, ally's y pos to center)
18          own_info (dict of tuple): Dict of tuple of (n, ): dict('
        agent_0', 'agent_1', ..., 'agent_N') Tuple of own information for
        each agent.
19                  ->own_info[agent_id]: each tuple contains information
         of (your health, your x position to center, your y position to
        center, last action you take, whether you are alived)
20      '''
21      available_move_actions = {}
22      enemy_info = {}
23      ally_info = {}
24      own_info = {}
25      action_num = 6+m
26      for id, obs in enumerate(global_state):
27          agent_id = f"agent_{id}"
28          offset = 0
29          al_ids = [f"agent_{al_id}" for al_id in range(n) if f"agent_{
        al_id}" != agent_id]
30          ally_info[agent_id] = {}
31          for al_id in al_ids:
32              ally_info[agent_id][al_id] = []
33              # whether the ally is visible or in the sight range of the
        agent
34              is_current_ally_visible = obs[offset: offset + 1]
35              ally_info[agent_id][al_id].append(is_current_ally_visible)
36              offset += 1
37              # distance to the ally
38              dist_to_ally = obs[offset: offset + 1]
39              ally_info[agent_id][al_id].append(dist_to_ally)
40              offset += 1
41              # ally's position relative to the agent
42              pos_x_to_ally = obs[offset: offset + 1]
43              ally_info[agent_id][al_id].append(pos_x_to_ally)
44              pos_y_to_ally = obs[offset + 1: offset + 2]
45              ally_info[agent_id][al_id].append(pos_y_to_ally)
46              offset += 2
47              # the time left for the ally to use the weapon
48              weapon_cooldown = obs[offset: offset + 1]
49              ally_info[agent_id][al_id].append(weapon_cooldown)
50              offset += 1
51              # health of the ally(0 to 1)
52              ally_health = obs[offset: offset + 1]
53              ally_info[agent_id][al_id].append(ally_health)
54              offset += 1
55              # ally's position relative to the center of the map
56              pos_x_to_center = obs[offset: offset + 1]
57              ally_info[agent_id][al_id].append(pos_x_to_center)
58              offset += 1
59              pos_y_to_center = obs[offset: offset + 1]
60              ally_info[agent_id][al_id].append(pos_y_to_center)
61              offset += 1
62              # the last action of the ally(str)
63              last_action = process_actions(obs[offset: offset + action_num
        ])
64              ally_info[agent_id][al_id].append(last_action)
65              offset += action_num
66              # whether the ally is alived
67              ally_alived = True
68              if last_action == "no operation":
69                  ally_alived = False
70              ally_info[agent_id][al_id].append(ally_alived)
71              ally_info[agent_id][al_id] = tuple(ally_info[agent_id][al_id
        ])
72          e_ids = [f"enemy_{e_id}" for e_id in range(m)]
```

```
 73         enemy_info[agent_id] = {}
 74         for e_id in e_ids:
 75             # whether the enemy is available to attack
 76             is_current_enemy_available_to_attack = obs[offset: offset +
    1]
 77             offset += 1
 78             # distance to the enemy
 79             dist_to_enemy = obs[offset: offset + 1]
 80             offset += 1
 81             # enemy's position relative to the agent
 82             pos_x_to_enemy = obs[offset: offset + 1]
 83             pos_y_to_enemy = obs[offset + 1: offset + 2]
 84             offset += 2
 85             # whether the enemy is visible or in the sight range of the
    agent
 86             is_current_enemy_visible = obs[offset: offset + 1]
 87             offset += 1
 88             # health of the enemy(0 to 1)
 89             enemy_health = obs[offset: offset + 1]
 90             offset += 1
 91             # enemy's position relative to the center of the map
 92             enemy_pos_x_to_center = obs[offset: offset + 1]
 93             offset += 1
 94             enemy_pos_y_to_center = obs[offset: offset + 1]
 95             offset += 1
 96             enemy_info[agent_id][e_id] = (
 97                 is_current_enemy_available_to_attack, dist_to_enemy,
    pos_x_to_enemy, pos_y_to_enemy,
 98                 is_current_enemy_visible, enemy_health,
    enemy_pos_x_to_center, enemy_pos_y_to_center)
 99
100         move_feat = obs[: 4]
101         available_moves= []
102         if move_feat[0] == 1:
103             available_moves.append("North")
104         if move_feat[1] == 1:
105             available_moves.append("South")
106         if move_feat[2] == 1:
107             available_moves.append("East")
108         if move_feat[3] == 1:
109             available_moves.append("West")
110         available_move_actions[agent_id] = available_moves
111         offset += 4
112
113         offset += 4
114         own_info[agent_id] = []
115         own_health = obs[offset: offset + 1]
116         own_info[agent_id].append(own_health)
117         offset += 1
118         own_pos_x_to_center = obs[offset: offset + 1]
119         own_info[agent_id].append(own_pos_x_to_center)
120         offset += 1
121         own_pos_y_to_center = obs[offset: offset + 1]
122         own_info[agent_id].append(own_pos_y_to_center)
123         offset += 1
124         own_last_action = process_actions(obs[offset: offset + action_num
    ])
125         own_info[agent_id].append(own_last_action)
126         offset += action_num
127         own_alived = True
128         if own_last_action == "no operation":
129             own_alived = False
130         own_info[agent_id].append(own_alived)
131         own_info[agent_id] = tuple(own_info[agent_id])
132
```

```
133         processed_global_state = (available_move_actions, enemy_info,
        ally_info, own_info)

134
135     return processed_global_state
```

**SMAC 2s vs 1sc map**

```
1  def process_global_state(observations, n=2, m=1):
2      '''
3      Param:
4          observation:
5                              Dict of list of (n, ): dict('agent_0', 'agent_1',
        ..., 'agent_N')
6                              List:
7                              Agent : (m, ) list of observation components
8          n: int, number of agents
9          m: int, number of enemies
10     Return:
11         obs (tuples of dict): Tuples of dict of (n, ): Tuple of each
        observation components processed from each agent's perspective by
        function "process_observation":
12             move_feats (dict of list): Dict of list of (n, ): dict('
        agent_0', 'agent_1', ..., 'agent_N') List of available moves for each
         agent.
13             enemy_info (dict of dict of tuple): Dict of dict of tuple of
        (n, ): dict('agent_0', 'agent_1', ..., 'agent_N') Tuple of m enemies
        information(enemy_0 to enemy_m) for each agent.
14             ally_info (dict of dict of tuple): Dict of dict of tuple of (
        n, ): dict('agent_0', 'agent_1', ..., 'agent_N') Tuple of n-1 ally
        information(exclude self) for each agent.
15             own_info (dict of tuple): Dict of tuple of (n, ): dict('
        agent_0', 'agent_1', ..., 'agent_N') Tuple of own information for
        each agent.
16     '''
17     move_feats = {}
18     enemy_info = {}
19     ally_info = {}
20     own_info = {}
21     action_num = 6+m
22     for id, obs in enumerate(observations):
23         agent_id = f"agent_{id}"
24         offset = 0
25         al_ids = [f"agent_{al_id}" for al_id in range(n) if f"agent_{
        al_id}" != agent_id]
26         ally_info[agent_id] = {}
27         for al_id in al_ids:
28             # whether the ally is visible or in the sight range of the
        agent
29             is_current_ally_visible = obs[offset: offset + 1]
30             offset += 1
31             # distance to the ally
32             dist_to_ally = obs[offset: offset + 1]
33             offset += 1
34             # ally's position relative to the agent
35             pos_x_to_ally = obs[offset: offset + 1]
36             pos_y_to_ally = obs[offset + 1: offset + 2]
37             offset += 2
38             # the time left for the ally to use the weapon
39             weapon_cooldown = obs[offset: offset + 1]
40             offset += 1
41             # health of the ally(0 to 1)
42             ally_health = obs[offset: offset + 1]
43             offset += 1
44             # shield of the ally(0 to 1)
```

```
45          ally_shield = obs[offset: offset + 1]
46          offset += 1
47          # ally's position relative to the center of the map
48          pos_x_to_center = obs[offset: offset + 1]
49          offset += 1
50          pos_y_to_center = obs[offset: offset + 1]
51          offset += 1
52          # the last action of the ally(str)
53          last_action = process_actions(obs[offset: offset + action_num
    ])
54          offset += action_num
55          # whether the ally is alived
56          ally_alived = True
57          if last_action == "no operation":
58              ally_alived = False
59          ally_info[agent_id][al_id] = (is_current_ally_visible,
    dist_to_ally, pos_x_to_ally, pos_y_to_ally,
60                                        weapon_cooldown, ally_health,
    ally_shield, pos_x_to_center, pos_y_to_center,
61                                        last_action, ally_alived)
62      e_ids = [f"enemy_{e_id}" for e_id in range(m)]
63      enemy_info[agent_id] = {}
64      for e_id in e_ids:
65          # whether the enemy is available to attack
66          is_current_enemy_available_to_attack = obs[offset: offset +
    1]
67          offset += 1
68          # distance to the enemy
69          dist_to_enemy = obs[offset: offset + 1]
70          offset += 1
71          # enemy's position relative to the agent
72          pos_x_to_enemy = obs[offset: offset + 1]
73          pos_y_to_enemy = obs[offset + 1: offset + 2]
74          offset += 2
75          # whether the enemy is visible or in the sight range of the
    agent
76          is_current_enemy_visible = obs[offset: offset + 1]
77          offset += 1
78          # health of the enemy(0 to 1)
79          enemy_health = obs[offset: offset + 1]
80          offset += 1
81          # enemy's position relative to the center of the map
82          enemy_pos_x_to_center = obs[offset: offset + 1]
83          offset += 1
84          enemy_pos_y_to_center = obs[offset: offset + 1]
85          offset += 1
86          enemy_info[agent_id][e_id] = (
87              is_current_enemy_available_to_attack, dist_to_enemy,
    pos_x_to_enemy, pos_y_to_enemy,
88              is_current_enemy_visible, enemy_health,
    enemy_pos_x_to_center, enemy_pos_y_to_center)
89
90      move_feat = obs[: 4]
91      available_moves= []
92      if move_feat[0] == 1:
93          available_moves.append("North")
94      if move_feat[1] == 1:
95          available_moves.append("South")
96      if move_feat[2] == 1:
97          available_moves.append("East")
98      if move_feat[3] == 1:
99          available_moves.append("West")
100     move_feats[agent_id] = available_moves
101     offset += 4
102
```

```
103        offset += 4
104        own_health = obs[offset: offset + 1]
105        offset += 1
106        own_shield = obs[offset: offset + 1]
107        offset += 1
108        own_pos_x_to_center = obs[offset: offset + 1]
109        offset += 1
110        own_pos_y_to_center = obs[offset: offset + 1]
111        offset += 1
112        own_last_action = process_actions(obs[offset: offset + action_num
    ])
113        offset += action_num
114        own_alived = True
115        if own_last_action == "no operation":
116            own_alived = False
117        own_info[agent_id] = (own_health, own_shield, own_pos_x_to_center
    , own_pos_y_to_center,
118                            own_last_action, own_alived)
119        obs = (move_feats, enemy_info, ally_info, own_info)
120    return obs
```

**SMAC 2c vs 64zg map**

```
1  def process_global_state(observations, n=2, m=64):
2      '''
3      Param:
4          observation:
5                          Dict of list of (n, ): dict('agent_0', 'agent_1',
     ..., 'agent_N')
6                          List:
7                          Agent : (m, ) list of observation components
8      n: int, number of agents
9      m: int, number of enemies
10     Return:
11         obs (tuples of dict): Tuples of dict of (n, ): Tuple of each
    observation components processed from each agent's perspective by
    function "process_observation":
12             move_feats (dict of list): Dict of list of (n, ): dict('
    agent_0', 'agent_1', ..., 'agent_N') List of available moves for each
     agent.
13             enemy_info (dict of dict of tuple): Dict of dict of tuple of
    (n, ): dict('agent_0', 'agent_1', ..., 'agent_N') Tuple of m enemies
    information(enemy_0 to enemy_m) for each agent.
14             ally_info (dict of dict of tuple): Dict of dict of tuple of (
    n, ): dict('agent_0', 'agent_1', ..., 'agent_N') Tuple of n-1 ally
    information(exclude self) for each agent.
15             own_info (dict of tuple): Dict of tuple of (n, ): dict('
    agent_0', 'agent_1', ..., 'agent_N') Tuple of own information for
    each agent.
16     '''
17     move_feats = {}
18     enemy_info = {}
19     ally_info = {}
20     own_info = {}
21     action_num = 6+m
22     for id, obs in enumerate(observations):
23         agent_id = f"agent_{id}"
24         offset = 0
25         al_ids = [f"agent_{al_id}" for al_id in range(n) if f"agent_{
    al_id}" != agent_id]
26         ally_info[agent_id] = {}
27         for al_id in al_ids:
28             # whether the ally is visible or in the sight range of the
    agent
```

```
29          is_current_ally_visible = obs[offset: offset + 1]
30          offset += 1
31          # distance to the ally
32          dist_to_ally = obs[offset: offset + 1]
33          offset += 1
34          # ally's position relative to the agent
35          pos_x_to_ally = obs[offset: offset + 1]
36          pos_y_to_ally = obs[offset + 1: offset + 2]
37          offset += 2
38          # the time left for the ally to use the weapon
39          weapon_cooldown = obs[offset: offset + 1]
40          offset += 1
41          # health of the ally(0 to 1)
42          ally_health = obs[offset: offset + 1]
43          offset += 1
44          # shield of the ally(0 to 1)
45          ally_shield = obs[offset: offset + 1]
46          offset += 1
47          # ally's position relative to the center of the map
48          pos_x_to_center = obs[offset: offset + 1]
49          offset += 1
50          pos_y_to_center = obs[offset: offset + 1]
51          offset += 1
52          # the last action of the ally(str)
53          last_action = process_actions(obs[offset: offset + action_num
    ])
54          offset += action_num
55          # whether the ally is alived
56          ally_alived = True
57          if last_action == "no operation":
58              ally_alived = False
59          ally_info[agent_id][al_id] = (is_current_ally_visible,
    dist_to_ally, pos_x_to_ally, pos_y_to_ally,
60                                          weapon_cooldown, ally_health,
    ally_shield, pos_x_to_center, pos_y_to_center,
61                                          last_action, ally_alived)
62      e_ids = [f"enemy_{e_id}" for e_id in range(m)]
63      enemy_info[agent_id] = {}
64      for e_id in e_ids:
65          # whether the enemy is available to attack
66          is_current_enemy_available_to_attack = obs[offset: offset +
    1]
67          offset += 1
68          # distance to the enemy
69          dist_to_enemy = obs[offset: offset + 1]
70          offset += 1
71          # enemy's position relative to the agent
72          pos_x_to_enemy = obs[offset: offset + 1]
73          pos_y_to_enemy = obs[offset + 1: offset + 2]
74          offset += 2
75          # whether the enemy is visible or in the sight range of the
    agent
76          is_current_enemy_visible = obs[offset: offset + 1]
77          offset += 1
78          # health of the enemy(0 to 1)
79          enemy_health = obs[offset: offset + 1]
80          offset += 1
81          # enemy's position relative to the center of the map
82          enemy_pos_x_to_center = obs[offset: offset + 1]
83          offset += 1
84          enemy_pos_y_to_center = obs[offset: offset + 1]
85          offset += 1
86          enemy_info[agent_id][e_id] = (
87              is_current_enemy_available_to_attack, dist_to_enemy,
    pos_x_to_enemy, pos_y_to_enemy,
```

```
88              is_current_enemy_visible, enemy_health,
        enemy_pos_x_to_center, enemy_pos_y_to_center)
89
90      move_feat = obs[: 4]
91      available_moves= []
92      if move_feat[0] == 1:
93          available_moves.append("North")
94      if move_feat[1] == 1:
95          available_moves.append("South")
96      if move_feat[2] == 1:
97          available_moves.append("East")
98      if move_feat[3] == 1:
99          available_moves.append("West")
100     move_feats[agent_id] = available_moves
101     offset += 4
102
103     offset += 4
104     own_health = obs[offset: offset + 1]
105     offset += 1
106     own_shield = obs[offset: offset + 1]
107     offset += 1
108     own_pos_x_to_center = obs[offset: offset + 1]
109     offset += 1
110     own_pos_y_to_center = obs[offset: offset + 1]
111     offset += 1
112     own_last_action = process_actions(obs[offset: offset + action_num
    ])
113     offset += action_num
114     own_alived = True
115     if own_last_action == "no operation":
116         own_alived = False
117     own_info[agent_id] = (own_health, own_shield, own_pos_x_to_center
    , own_pos_y_to_center,
118                 own_last_action, own_alived)
119     obs = (move_feats, enemy_info, ally_info, own_info)
120 return obs
```

**Sample output**
We provide some sample strategy generated by LLM for the simple scenerios.

Based on the environment description and game rules, I'll provide a strategy focused on an aggressive approach to eliminate all enemy units within the 60-step limit. Here's a concise, clear, and actionable set of instructions:

1. Goal: Eliminate all 3 enemy Marine units within 60 steps.

2. Problem: We need to overcome the symmetrical nature of the battle (3v3 Marines) and exploit any positional advantages to win quickly.

3. Approach: Adopt an aggressive strategy with coordinated attacks to focus fire and over-whelm enemies one by one.

4. Observations and Task Breakdown:

a) Initial Positioning (Steps 1-5): - All agents: Move to form a tight triangle formation, re-ducing exposure and enabling focused fire. - Agent 1: Move forward slightly to act as bait. - Agents 2 and 3: Position slightly behind Agent 1 on either side.

b) Engagement (Steps 6-20): - Agent 1: Move forward to draw enemy fire. - Agents 2 and 3: Attack the nearest enemy unit, focusing fire on the same target. - If Agent 1 takes heavy damage, it should retreat behind Agents 2 and 3.

c) Focus Fire (Steps 21-40): - All agents: Continuously attack the same enemy unit until it's eliminated. - Prioritize the enemy unit with the lowest health. - If an enemy unit tries to retreat, pursue it while maintaining formation.

d) Adapt and Eliminate (Steps 41-60): - After eliminating the first enemy, quickly shift focus to the next nearest enemy. - If any agent's health drops below 30 percent, it should retreat and use the other agents as cover. - Maintain aggressive pursuit of enemy units, don't allow them to regroup or heal.

Throughout the engagement: - Utilize the maximum shooting range of 6 to attack enemies while minimizing damage taken. - Use the sight range of 9 to spot enemies early and plan movements accordingly. - Constantly reassess the battlefield and adjust tactics if the initial approach isn't working. - If victory isn't achieved by step 50, take calculated risks to ensure all enemies are eliminated before the 60-step limit.

This strategy emphasizes aggressive play, coordinated attacks, and adaptability to overcome the symmetrical nature of the battle and achieve victory within the time limit.

C.3    MORE PROMPTS FOR PLANNING FUNCTION GENERATION

To ensure that the function generation integrates effectively with the reinforcement learning (RL) training, we provide a detailed prompt that contains the goals and expected formats of the functions. This prompt is designed to guide the creation of the functions, ensuring they enhance agent cooperation and coordination. The prompt is as follows:

Your task is to create a planning function and a reward function that work together to improve agent cooperation. The planning function should help each agent reach its goal, and the reward function should encourage smooth collaboration. Both functions should follow the guide from tips and focus on ensuring that the agents coordinate their movements to reach their goals simultaneously.

The environment code information is provided as follows:

def process_global_state(global_state, n=3, m=3):
...
return processed_global_state

The format for function generation is as follows:

The planning function should look like:

def planning_function(processed_global_state, available_actions):
"""
Determines optimal tasks for each agent based on the current battle state.
Args:
processed_global_state: A tuple containing (available_move_actions, enemy_info, ally_info, own_info)
available_actions: A dict of available action indices for each agent

Returns:
llm_tasks: Dict containing optimal tasks for each agent (Assignment Class)
"""
...

return llm_tasks

The returned 'll_tasks' should be in the ¡tasks assignment class¿ as specified. Use 'processed_global_state' to inform decision-making.

The reward function should look like:

def compute_reward(processed_global_state, llm_tasks, tasks):
"""
Calculate rewards based on the tasks assigned and their outcomes.

Args:
processed_global_state: Returned from the function process_global_state(global_state, n, m)
llm_tasks (dict): Dictionary containing tasks assigned to each agent.
tasks (dict): Dictionary of tasks actually performed by each agent, e.g., 'agent_0': ...

Returns:
reward: Dict containing rewards for each agent. For example: 'agent_0': reward1, 'agent_1': reward2, ...
"""

...

return reward

You should adjust the reward value for each component based on the importance as suggested in the tips.
You may use or import any necessary APIs for code generation, but do not write into a class object.
The generated functions should only include 'planning_function' and 'compute_reward'. Do not create new variables or subfunctions.
Strictly follow the size, shape, and format of the action space and 'processed_global_state'.
Think step-by-step before generating the two functions based on the information provided. First, consider the information available in 'processed_global_state' and how to use it in the functions. Second, analyze the environment description and determine the appropriate strategies and task assignments for each agent in this scenario.
Ensure the functions not only work correctly but also maximize agent coordination based on the instructions.

By supplying this prompt, we aim to generate functions that not only operate correctly within the RL framework but also maximize agent coordination based on the provided instructions. This approach ensures that the agents learn to work together effectively, ultimately enhancing the overall performance of the multi-agent system.

## D    EXAMPLES OF GENERATED PLANNING FUNCTIONS

```
import numpy as np

def planning_function(processed_state):
    """
    Determines optimal tasks for each agent based on the current state.

    Args:
        processed_state: A tuple containing food location and level,
    agent position and level.

    Returns:
```

```
            dict: Optimal tasks for each agent ('No op','Target food 0','
    Target food 1','Pickup')
        """
        food_info, agents_info = processed_state
        llm_tasks = {}

        # Find available food items
        available_food = [f for f, info in food_info.items() if info is not
    None]

        if not available_food:
            return {agent: 'No op' for agent in agents_info}

        # Calculate distances to food items
        distances = {}
        for food in available_food:
            food_pos = food_info[food][0]
            food_level = food_info[food][1]
            for agent, (agent_pos, agent_level) in agents_info.items():
                dist = np.linalg.norm(np.array(food_pos) - np.array(agent_pos
    ))
                if food not in distances or dist < distances[food][1]:
                    distances[food] = (agent, dist)

        # Sort food by distance
        sorted_food = sorted(distances.items(), key=lambda x: x[1][1])

        # Assign tasks
        target_food = sorted_food[0][0]
        food_pos = food_info[target_food][0]
        food_level = food_info[target_food][1]

        total_agent_level = sum(level for _, level in agents_info.values())

        for agent in agents_info:
            agent_pos, _ = agents_info[agent]
            if np.linalg.norm(np.array(food_pos) - np.array(agent_pos)) <= 1
    and total_agent_level >= food_level:
                llm_tasks[agent] = 'Pickup'
            else:
                llm_tasks[agent] = f'Target {target_food}'

        return llm_tasks
```

## E   REWARD GENERATION WITH FEEDBACK

The functions provided here are the generated reward function including: iteration 1, iteration 2, iteration 3, iteration 4.

Iteration 1:

```
def compute_reward(processed_state, actions):
    """
    Calculate rewards based on the tasks assigned and their outcomes.

    Args:
        processed_state: returned from function process_state(state, p, f
    )
        actions (dict): dictionary of a integer action that actually
    perform by each agent. E.g. {"agent_0": 2, "agent_1": 4, ...}

    Returns:
        reward: Dict containing rewards for each agent. For example: {'
    agent_0': reward1, 'agent_1', reward2, ...}
```

```
11      """
12      food_info, agents_info = processed_state
13      reward = {agent_id: 0 for agent_id in agents_info.keys()}
14
15      # Reward for picking up food
16      pickup_agents = [agent_id for agent_id, action in actions.items() if
        action == 5]
17      if pickup_agents:
18          food_positions = [food[0] for food in food_info.values() if food
        is not None]
19          for agent_id in pickup_agents:
20              agent_pos = agents_info[agent_id][0]
21              if any(abs(agent_pos[0] - food_pos[0]) + abs(agent_pos[1] -
        food_pos[1]) <= 1 for food_pos in food_positions):
22                  reward[agent_id] += 10  # Reward for attempting pickup
        near food
23
24      # Reward for moving towards food
25      for agent_id, action in actions.items():
26          if action in [1, 2, 3, 4]:  # Moving actions
27              agent_pos = agents_info[agent_id][0]
28              closest_food = min((food for food in food_info.values() if
        food is not None),
29                                 key=lambda f: abs(agent_pos[0] - f[0][0])
        + abs(agent_pos[1] - f[0][1]),
30                                 default=None)
31              if closest_food:
32                  old_distance = abs(agent_pos[0] - closest_food[0][0]) +
        abs(agent_pos[1] - closest_food[0][1])
33                  new_pos = list(agent_pos)
34                  if action == 1: new_pos[0] -= 1
35                  elif action == 2: new_pos[0] += 1
36                  elif action == 3: new_pos[1] -= 1
37                  elif action == 4: new_pos[1] += 1
38                  new_distance = abs(new_pos[0] - closest_food[0][0]) + abs
        (new_pos[1] - closest_food[0][1])
39                  if new_distance < old_distance:
40                      reward[agent_id] += 1  # Reward for moving closer to
        food
41
42      # Reward for agent proximity (encouraging potential collaboration)
43      agent_positions = [info[0] for info in agents_info.values()]
44      if len(agent_positions) > 1:
45          distance = abs(agent_positions[0][0] - agent_positions[1][0]) +
        abs(agent_positions[0][1] - agent_positions[1][1])
46          proximity_reward = max(0, (5 - distance) / 2)  # Max reward when
        adjacent, decreasing with distance
47          for agent_id in reward:
48              reward[agent_id] += proximity_reward
49
50      return reward
```

Iteration 2:

```
1  def compute_reward(processed_state, actions):
2      """
3      Calculate rewards based on the tasks assigned and their outcomes.
4
5      Args:
6          processed_state: returned from function process_state(state, p, f
        )
7          actions (dict): dictionary of a integer action that actually
        perform by each agent. E.g. {"agent_0": 2, "agent_1": 4, ...}
8
9      Returns:
```

```
10          reward: Dict containing rewards for each agent. For example: {'
            agent_0': reward1, 'agent_1', reward2, ...}
11      """
12      food_info, agents_info = processed_state
13      reward = {agent_id: 0 for agent_id in agents_info.keys()}
14
15      # Reward for picking up food
16      pickup_agents = [agent_id for agent_id, action in actions.items() if
            action == 5]
17      if len(pickup_agents) == len(agents_info):  # All agents attempting
            pickup
18          food_positions = [food[0] for food in food_info.values() if food
                is not None]
19          if all(any(abs(agents_info[agent_id][0][0] - food_pos[0]) + abs(
                agents_info[agent_id][0][1] - food_pos[1]) <= 1 for food_pos in
                food_positions) for agent_id in pickup_agents):
20              for agent_id in pickup_agents:
21                  reward[agent_id] += 50  # Higher reward for coordinated
                        pickup
22
23      # Reward for moving towards food and staying close to other agents
24      for agent_id, action in actions.items():
25          if action in [1, 2, 3, 4]:  # Moving actions
26              agent_pos = agents_info[agent_id][0]
27              closest_food = min((food for food in food_info.values() if
                    food is not None),
28                              key=lambda f: abs(agent_pos[0] - f[0][0])
                    + abs(agent_pos[1] - f[0][1]),
29                              default=None)
30              if closest_food:
31                  old_distance = abs(agent_pos[0] - closest_food[0][0]) +
                        abs(agent_pos[1] - closest_food[0][1])
32                  new_pos = list(agent_pos)
33                  if action == 1: new_pos[0] -= 1
34                  elif action == 2: new_pos[0] += 1
35                  elif action == 3: new_pos[1] -= 1
36                  elif action == 4: new_pos[1] += 1
37                  new_distance = abs(new_pos[0] - closest_food[0][0]) + abs
                        (new_pos[1] - closest_food[0][1])
38                  if new_distance < old_distance:
39                      reward[agent_id] += 2  # Increased reward for moving
                            closer to food
40
41              # Reward for staying close to other agents
42              other_agents = [a for a in agents_info.keys() if a !=
                    agent_id]
43              for other_agent in other_agents:
44                  other_pos = agents_info[other_agent][0]
45                  old_agent_distance = abs(agent_pos[0] - other_pos[0]) +
                        abs(agent_pos[1] - other_pos[1])
46                  new_agent_distance = abs(new_pos[0] - other_pos[0]) + abs
                        (new_pos[1] - other_pos[1])
47                  if new_agent_distance <= old_agent_distance:
48                      reward[agent_id] += 1  # Reward for maintaining or
                            decreasing distance to other agents
49
50      # Penalty for no-op when food is available
51      for agent_id, action in actions.items():
52          if action == 0 and any(food is not None for food in food_info.
                values()):
53              reward[agent_id] -= 1  # Small penalty for no-op when food is
                 available
54
55      return reward
```

Iteration 3:

```
def compute_reward(processed_state, actions):
    """
    Calculate rewards based on the tasks assigned and their outcomes.

    Args:
        processed_state: returned from function process_state(state, p, f
)
        actions (dict): dictionary of a integer action that actually
perform by each agent. E.g. {"agent_0": 2, "agent_1": 4, ...}

    Returns:
        reward: Dict containing rewards for each agent. For example: {'
agent_0': reward1, 'agent_1', reward2, ...}
    """
    food_info, agents_info = processed_state
    reward = {agent_id: 0 for agent_id in agents_info.keys()}

    # Reward for picking up food
    pickup_agents = [agent_id for agent_id, action in actions.items() if
action == 5]
    if len(pickup_agents) == len(agents_info):  # All agents attempting
pickup
        food_positions = [food[0] for food in food_info.values() if food
is not None]
        if food_positions and all(any(abs(agents_info[agent_id][0][0] -
food_pos[0]) + abs(agents_info[agent_id][0][1] - food_pos[1]) <= 1
for food_pos in food_positions) for agent_id in pickup_agents):
            total_agent_level = sum(agents_info[agent_id][1] for agent_id
 in pickup_agents)
            food_level = max(food[1] for food in food_info.values() if
food is not None)
            if total_agent_level >= food_level:
                for agent_id in pickup_agents:
                    reward[agent_id] += 100  # Higher reward for
successful coordinated pickup

    # Reward for moving towards food and staying close to other agents
    for agent_id, action in actions.items():
        if action in [1, 2, 3, 4]:  # Moving actions
            agent_pos = agents_info[agent_id][0]
            closest_food = min((food for food in food_info.values() if
food is not None),
                               key=lambda f: abs(agent_pos[0] - f[0][0])
+ abs(agent_pos[1] - f[0][1]),
                               default=None)
            if closest_food:
                old_distance = abs(agent_pos[0] - closest_food[0][0]) +
abs(agent_pos[1] - closest_food[0][1])
                new_pos = list(agent_pos)
                if action == 1: new_pos[0] -= 1
                elif action == 2: new_pos[0] += 1
                elif action == 3: new_pos[1] -= 1
                elif action == 4: new_pos[1] += 1
                new_distance = abs(new_pos[0] - closest_food[0][0]) + abs
(new_pos[1] - closest_food[0][1])
                if new_distance < old_distance:
                    reward[agent_id] += 5  # Increased reward for moving
closer to food

            # Reward for staying close to other agents
            other_agents = [a for a in agents_info.keys() if a !=
agent_id]
            for other_agent in other_agents:
```

```
47              other_pos = agents_info[other_agent][0]
48              old_agent_distance = abs(agent_pos[0] - other_pos[0]) +
    abs(agent_pos[1] - other_pos[1])
49              new_agent_distance = abs(new_pos[0] - other_pos[0]) + abs
    (new_pos[1] - other_pos[1])
50              if new_agent_distance < old_agent_distance:
51                  reward[agent_id] += 3  # Increased reward for
    decreasing distance to other agents
52
53      # Penalty for no-op when food is available
54      for agent_id, action in actions.items():
55          if action == 0 and any(food is not None for food in food_info.
    values()):
56              reward[agent_id] -= 5  # Increased penalty for no-op when
    food is available
57
58      return reward
```

Iteration 4:

```
1  def compute_reward(processed_state, actions):
2      """
3      Calculate rewards based on the tasks assigned and their outcomes.
4
5      Args:
6          processed_state: returned from function process_state(state, p, f
    )
7          actions (dict): dictionary of a integer action that actually
    perform by each agent. E.g. {"agent_0": 2, "agent_1": 4, ...}
8
9      Returns:
10          reward: Dict containing rewards for each agent. For example: {'
    agent_0': reward1, 'agent_1', reward2, ...}
11      """
12      food_info, agents_info = processed_state
13      reward = {agent_id: 0 for agent_id in agents_info.keys()}
14
15      # Reward for picking up food
16      pickup_agents = [agent_id for agent_id, action in actions.items() if
    action == 5]
17      if len(pickup_agents) == len(agents_info):  # All agents attempting
    pickup
18          food_positions = [food[0] for food in food_info.values() if food
    is not None]
19          if food_positions and all(any(abs(agents_info[agent_id][0][0] -
    food_pos[0]) + abs(agents_info[agent_id][0][1] - food_pos[1]) <= 1
    for food_pos in food_positions) for agent_id in pickup_agents):
20              total_agent_level = sum(agents_info[agent_id][1] for agent_id
     in pickup_agents)
21              food_level = max(food[1] for food in food_info.values() if
    food is not None)
22              if total_agent_level >= food_level:
23                  for agent_id in pickup_agents:
24                      reward[agent_id] += 200  # Higher reward for
    successful coordinated pickup
25
26      # Reward for moving towards food and staying close to other agents
27      for agent_id, action in actions.items():
28          if action in [1, 2, 3, 4]:  # Moving actions
29              agent_pos = agents_info[agent_id][0]
30              closest_food = min((food for food in food_info.values() if
    food is not None),
31                                 key=lambda f: abs(agent_pos[0] - f[0][0])
    + abs(agent_pos[1] - f[0][1]),
32                                 default=None)
```

```
33          if closest_food:
34              old_distance = abs(agent_pos[0] - closest_food[0][0]) +
        abs(agent_pos[1] - closest_food[0][1])
35              new_pos = list(agent_pos)
36              if action == 1: new_pos[0] -= 1
37              elif action == 2: new_pos[0] += 1
38              elif action == 3: new_pos[1] -= 1
39              elif action == 4: new_pos[1] += 1
40              new_distance = abs(new_pos[0] - closest_food[0][0]) + abs
        (new_pos[1] - closest_food[0][1])
41              if new_distance < old_distance:
42                  reward[agent_id] += 10  # Increased reward for moving
         closer to food
43
44          # Reward for staying close to other agents
45          other_agents = [a for a in agents_info.keys() if a !=
        agent_id]
46          for other_agent in other_agents:
47              other_pos = agents_info[other_agent][0]
48              old_agent_distance = abs(agent_pos[0] - other_pos[0]) +
        abs(agent_pos[1] - other_pos[1])
49              new_agent_distance = abs(new_pos[0] - other_pos[0]) + abs
        (new_pos[1] - other_pos[1])
50              if new_agent_distance < old_agent_distance:
51                  reward[agent_id] += 5  # Increased reward for
        decreasing distance to other agents
52
53      # Penalty for no-op when food is available
54      for agent_id, action in actions.items():
55          if action == 0 and any(food is not None for food in food_info.
        values()):
56              reward[agent_id] -= 10  # Increased penalty for no-op when
        food is available
57
58      # Completion bonus
59      if all(food is None for food in food_info.values()):
60          for agent_id in agents_info:
61              reward[agent_id] += 500  # Large bonus for completing the
        task
62
63      return reward
```

