# OpenReview forum: "YOLO-MARL: You Only LLM Once for Multi-agent Reinforcement Learning"
_ICLR.cc/2025/Conference — Submitted to ICLR 2025_

### Official Review · Reviewer_mMPd · 2024-10-26

**Soundness:** 3
**Presentation:** 2
**Contribution:** 2
**Rating:** 5
**Confidence:** 3

**Summary:**

The paper proposes to use LLMs to assist in the training of multiagent Reinforcement Learning tasks, by letting the LLM generate Strategy and Planning functions, those functions are used to modify the MDP for better training.

**Strengths:**

- Contemporary paper proposing to use LLMs to better design the MDP description.
- LLM-guided multiagent RL is a novel subject
- Authors were considerate about the explosive computational cots of using the LLM at every step the agents would take and thus engineered functions in a way that the LLM would need to be queried only once for the environment design.

**Weaknesses:**

- The main issue I see is that, although the LLM is used to assist in the design of the MDP description, a great deal of domain knowledge is still required of the designer. Put together, the work to format the environment description for the strategy generation and state interpretation steps is on par or even harder than more classical approaches of performing feature engineering such as OO-MDPs or simply  dedicating a good effort into modeling the task well.

- While the experimental results show significantly better results for the proposed approach, I feel it's an unfair comparison due to the significant effort required to describe well the environment to the LLM. The main argument in favor of the approach is that it required less coding when compared to classical feature engineering, but overall the same mental exercise would be needed not to mention the additional costs associated to using the LLM.

- My suggestion to make the approach more general and impactful is to instead use a multi-modal foundation model to handcraft some fixed prompts (that would be general for any environment) to analyze the environment autonomously through raw observations (such as images). That way, the approach would require significantly less effort from the designer and would make the approach more impactful.

- For the reasons above, in my opinion the paper does not reach the impact level of a main track ICLR paper.

- minor: I would name the approach in another way given there is already a widely famous YOLO model

**Questions:**

no specific question.

---

> ### Author Response · Authors · 2024-11-23
>
> For weakness 1 and 2:
> So the description for each environment can be directly found on the environments’ website(github for example). We simply copy and paste them to feed into the LLM. And for your concern, this is also why we use LLM to generate strategies for itself instead of writing the strategies by our own. This means that for a brand new environment, all you need to do is to go to the website, copy and paste the description and then let LLM do all the work.
>
> For weakness 3:
> Thank you for your insightful suggestion regarding the use of multi-modal foundation models to autonomously analyze environments through raw observations like images. We agree that this approach has the potential to make solutions more general and impactful by reducing the effort required from designers.
>
> In our current pipeline, the primary objective is to generate Python functions, which demands the advanced code generation capabilities of large language models (LLMs). We did consider incorporating multi-modal foundation models at the beginning of our project. However, at the time of our research, leading models such as GPT-4 and Claude-3.5 did not support image inputs, limiting our ability to process raw visual observations directly.
>
> Additionally, relying on image inputs would necessitate interacting with the foundation model at every step to continuously analyze the environment. This frequent interaction could introduce significant computational overhead and latency, which runs counter to our goal of developing an efficient and streamlined pipeline.
>
> We wholeheartedly believe that multi-modal foundation models represent a promising direction for future research. As these models continue to advance and better support image inputs, we are excited about the possibilities they offer and plan to explore their integration in our future work.

---

> > ### Comment · Reviewer_mMPd · 2024-11-26
> >
> > Just a small comment. For any practical application there won't be a "github page" with an RL-like description of the environment, and this will have to be put together by the designer, which is almost the same amount of work than writing the code itself using a proper IDE.
> >
> > But overall, it seems like there was no gross misunderstanding on matters related to my scoring. THerefore I keep my recommendation.

---

> ### Author Response · Authors · 2024-11-26
>
> Thanks for your reply.
>
> 1. Regarding the environment description, we want to clarify that the input required for our framework is simple and straightforward. Details about the prompt can be found in Appendix C. For each environment, the description consists of only one or two sentences. Even without prior knowledge of a new environment, users can easily create a brief description using common sense.
>
> 2. The following previously published works also rely on some form of prior knowledge or information available on the internet as initial prompts for LLMs:
>
>     a. **EUREKA[1]**: Utilizes unmodified environment source code and language task descriptions as context to zero-shot generate executable reward functions using a coding LLM.
>
>     b. **TEXT2REWARD[2]**: Provides a compact representation in Pythonic style, including environment abstractions and functions that serve as background knowledge for the LLM.
>
>     c. **Reward Design with Language Models[3]**: Supplies the LLM with a task description, a user’s objective description, an episode outcome converted into a string, and a question asking if the episode outcome satisfies the user’s objective.
>
>    d. **ProgPrompt[4], Code as Policies[5], GENSIM[6]**: These works also require human involvement for prompting, such as few-shot prompting. They often include example trajectories within the prompt to illustrate how tasks should be completed.
> These works highlight that human labor is commonly required for designing prompts in prior works, whereas our framework minimizes this effort by simplifying the process.
>
> 3, The environments used in our study are based on “Benchmarking Multi-Agent Deep Reinforcement Learning Algorithms in Cooperative Tasks”[7]. Detailed descriptions of these environments are already provided in the original paper and its accompanying documentation, where available. Additionally, all other components, such as strategies, are automatically generated by the LLM, eliminating the need for users to design them manually—a process that can be time-consuming and often requires expert knowledge. As a result, our framework significantly reduces the effort and time needed for these tedious tasks.
>
> [1] Y. J. Ma et al., “EUREKA: HUMAN-LEVEL REWARD DESIGN VIA CODING LARGE LANGUAGE MODELS.”  arXiv preprint arXiv:2310.12931, 2024.
>
> [2] T. Xie et al., “Text2Reward: Automated Dense Reward Function Generation for Reinforcement Learning,” Sep. 21, 2023, arXiv: arXiv:2309.11489.
>
> [3] M. Kwon, S. M. Xie, K. Bullard, and D. Sadigh, “Reward Design with Language Models,” Feb. 27, 2023, arXiv: arXiv:2303.00001.
>
> [4] I. Singh et al., “ProgPrompt: Generating Situated Robot Task Plans using Large Language Models,” Sep. 22, 2022, arXiv: arXiv:2209.11302.
>
> [5] J. Liang et al., “Code as Policies: Language Model Programs for Embodied Control,” May 25, 2023, arXiv: arXiv:2209.07753.
>
> [6] L. Wang et al., “GenSim: Generating Robotic Simulation Tasks via Large Language Models,” Jan. 21, 2024, arXiv: arXiv:2310.01361.

---

### Official Review · Reviewer_WvRh · 2024-11-02

**Soundness:** 2
**Presentation:** 3
**Contribution:** 2
**Rating:** 5
**Confidence:** 4

**Summary:**

The authors propose a novel framework (YOLO-MARL) to enhance MARL by integrating LLMs for high-level task planning in cooperative game environments. The motivation is to minimize computational demands by using only a single LLM interaction per game. YOLO-MARL includes four modules - Strategy Generation, State Interpretation, Planning Function Generation, and MARL Training. Experiments were conducted in useful benchmarks like Level-Based Foraging, MPE, and SMAC, with ablation studies to identify the role of some of the components of the proposed approach.

**Strengths:**

- The proposed idea is new, interesting, and well-motivated.
- The paper is easy to read and follow.
- Ample supplementary material containing examples and code helps in better understanding the proposed approach.

**Weaknesses:**

- Some more experiments and analysis/explanations may be required.
- See questions.

**Questions:**

- Does table 1 refer to the same results in Figure 2? Can you mention the standard deviations in the table too? I see a high overlap in them for some results, which raises a concern about the quantifiable impact that using LLMs is making in the proposed approach.
- Can you consider training Qmix on LBF for longer (about 10M)? This would help observe/compare the performances till/after convergences. Both approaches still seem to not have converged yet.
- How many independent trials were done for experiments in Section 6?
- About experiments on smac, can you give more details on the planning function generation using LLMs in this environment, as the actions here are complicated when compared to MPE. Also, the results on smac are almost overlapping. So, does this suggest that we may not need YOLO-MARL on smac maps? Did you also run qmix and YOLO-MARL+Qmix on smac maps? Also, did you consider smacv2? Any comment on why or why not?
- For experiments on reward generation in your approach, does the LLM generate a reward function? When does the LLM generate it? Before training (just like the strategy and planning function generation)? I am also unsure of more details on how this reward was generated.
- In Figure 2, any comment on the dip in performance of MADDPG (and YOLO-MARL + MADDPG) around 450K episodes?
- Referring to Figures 3 and 4, do you think the scalability of the proposed approach is a concern? The results almost start overlapping on increasing the number of agents in MPE.
- Minor: Can you label the axes in figures.

---

> ### Author Response · Authors · 2024-11-23
>
> Apologies for any confusion. This work focuses on designing a framework that leverages LLM guidance to enhance the training of MARL algorithms, with the training process and curve influenced by the underlying algorithm. To account for variability across random seeds, we report the average mean reward to demonstrate the effectiveness of our framework. By incorporating our pipeline, MARL policies (e.g., MAPPO, MADDPG) outperform baselines with shorter training times and higher mean episode rewards, as shown in [insert specific improvement metric].
> | Method\Steps     | 0.2M Mean Return        | 0.4M Mean Return        | 1.5M Mean Return       | 2M Mean Return         |
> |-------------------|-------------------------|--------------------------|------------------------|------------------------|
> | **QMIX (MARL)**   | 0.00(±0.0023)          | 0.01(±0.014)            | 0.25(±0.072)           | 0.38(±0.15)           |
> | **QMIX (YOLO-MARL)** | **0.01(±0.0012)**    | **0.02(±0.03)**         | **0.60(±0.113)**       | **0.78(±0.104)**      |
> | **MADDPG (MARL)** | 0.09(±0.004)           | 0.33(±0.09)             | 0.26(±0.112)           | 0.32(±0.104)          |
> | **MADDPG (YOLO-MARL)** | **0.13(±0.0032)**  | **0.38(±0.062)**        | **0.39(±0.031)**       | **0.44(±0.042)**      |
> | **MAPPO (MARL)**  | 0.31(±0.00)            | 0.72(±0.124)            | 0.99(±0.001)           | 0.99(±0.00)           |
> | **MAPPO (YOLO-MARL)** | **0.93(±0.012)**    | **0.98(±0.002)**        | **0.99(±0.00)**        | **0.99(±0.00)**       |
>
> Regarding to reward generation, we would like to clarify that the ablation study on reward generation involves using the LLM to generate a reward function that replaces the original reward provided by the environment. The reward generation process is similar to the planning function generation process and occurs only once, during the initial use of a specific new environment as input prompt for the LLM. This stage happens before the MARL policy training begins. We have clarified this point in the revised paper with the following explanation: **“For reward generation without feedback, the reward function is generated at the same stage as the planning function to ensure a fair comparison. Specifically, the reward function is generated before the training process begins for each new environment. For reward generation with feedback, the initial reward function is generated in the same way as in the no-feedback scenario. However, the process then becomes iterative: we first complete a full training cycle on the environment, then provide the LLM with feedback based on the training performance. This feedback, combined with the previous prompts, is used to refine the initially generated reward function."**
>
> For 10M step, we would like to clarify that the steps we used in our experiments strictly adhere to the methodology outlined in **“Benchmarking Multi-Agent Deep Reinforcement Learning Algorithms in Cooperative Tasks.” **. Specifically, as their experiments were conducted in the LBF environment for 2M steps, we adopted the same setting to ensure a fair comparison with the original work. Given that the results clearly demonstrate our framework's superior performance compared to the baseline, we believe it is not necessary to extend the experiments to longer steps at this stage.
>
> Regarding SMACv2, we found its training time significantly longer than SMAC and encountered challenges in accessing core objects and modifying the environment wrapper interface. Integrating our pipeline required incorporating LLM-generated functions, such as the planning function or reward signal, into the training process, necessitating adjustments to components like the critic neural network. These complexities made SMACv2 less suitable for our framework, so we focused on SMAC, which offered a more flexible and accessible environment. This choice highlights why only certain MARL environments, like SMAC, were viable testbeds for our proposed work.
>
> As for the dip observed in YOLO-MARL, we believe it reflects a similar dip in the baseline, as our method builds on the underlying MARL algorithm. For example, MADDPG's inherent instability could contribute to this behavior. The baseline implementation was also directly adapted from **“Benchmarking Multi-Agent Deep Reinforcement Learning Algorithms in Cooperative Tasks” and its GitHub repository.** Since our framework focuses on leveraging LLM guidance to enhance MARL training, the dip is likely due to the training process and the intrinsic limitations of the base algorithm rather than our framework design.

---

> > ### Comment · Reviewer_WvRh · 2024-12-01
> > **More clarifications are required**
> >
> > Thank you for your response and efforts! While your clarifications addressed some aspects, several of the questions and concerns from my review remain partially unanswered or lack sufficient elaboration. Below, I summarize the key gaps that still remain and provide suggestions for improvement:
> >
> > 1. The reported standard deviations in Table 1 are surprisingly low (with some of them even being 0.0), particularly given the variability typically seen in MARL experiments. Can you provide more details on how these values were calculated and whether they reflect sufficient statistical rigor? This is crucial to validate the claims of performance improvements.
> >
> > 2. It is still unclear if Table 1 refers to the same results shown in Figure 2. The inclusion of standard deviations in Table 1 helps, but it would be useful to explicitly connect these results to Figure 2. The high overlap in results across methods raises questions about the quantifiable impact of using LLMs, which needs clearer explanation in the paper.
> >
> > 3. The number of independent trials conducted for the experiments in Section 6 is not explicitly mentioned. This is critical for understanding the robustness of those results and should be clearly stated.
> >
> > 4. Details on how the planning function was generated using LLMs in SMAC are still insufficient, especially given the complexity of actions in SMAC compared to MPE. I do not understand what the LLM instructions (template/intuition) to generate the planning function for SMAC actions (very complicated) were like. I am referring to defining an assignment planning function which was explained for MPE as an example. Similarly, how or what was done for SMAC, given that it has a much more complicated and large action space.
> >
> > 5. Your explanation about reward function generation clarifies that the LLM generates the reward function before training begins, but further details on how this reward was generated and whether it consistently aligns with task objectives are still missing (I would like to know this, especially for SMAC; was it restricted to being sparse such as “team win or lose”? Or what other instructions/template was followed for the LLM prompts).
> >
> > While your previous response addresses some concerns, clearer and more detailed answers to the questions above, particularly on low standard deviations, independent trials, and reward generation instructions, would significantly strengthen the paper. Thank you again for your thoughtful engagement.

---

> > > ### Author Response · Authors · 2024-12-02
> > >
> > > Thank you for your response and for pointing out these important questions. I apologize for any confusion caused earlier. We will make our GitHub repository public as soon as we clean up the code, enabling everyone to experiment with Claude or Chat-GPT at a cost of less than a dollar and gain insights into how LLMs work.
> > >
> > > Regarding Table 1 and Figure 2, the results in Table 1 are directly derived from Figure 2, as mentioned in the table caption: “The corresponding results can be found in Figure 2.” To compute the return means corresponding to the steps, we measured values approximately from Figure 2. You can find an example of the generated planning function in Appendix D. We strongly believe that the generated strategies and planning functions align with common human reasoning for winning the game.
> > >
> > > We appreciate your concerns regarding the overlap in results across methods and its implications for quantifying the impact of LLMs. The baseline methods we employed were directly adapted from EPyMARL [1], which offers well-tested MARL algorithms with high performance but inherent variability. This variability, which can significantly affect performance, also contributes to overlapping results. However, in our experiments, we controlled for this by using the same seed for both the baseline and YOLO-MARL comparisons. Under these controlled conditions, YOLO-MARL consistently outperformed the baseline. Additionally, as shown in the variance trends provided in the final section (calculated directly using wandb for clarity), performance variability depends significantly on the stochastic nature of training. Although overlapping results exist, all experiments were conducted fairly across three seeds, ensuring direct comparisons under the same conditions.
> > >
> > > In Section 6, we conducted one independent trial for each experiment. The primary reason for this limitation was the difficulty of generating planning functions without the strategy interpretation module. Without this module, LLM-generated code often lacked logical coherence or contained execution errors. As shown in Appendix B.3, Figure 10, these planning functions were frequently incomplete or buggy, leading to poor performance. Without incorporating the strategy interpretation module into the LLM’s prompts or providing environment source code as doing in Eureka[2] to align input and output correctly, the generated code had a high likelihood of errors or logical flaws. Consequently, the planning functions generated without this module exhibited inferior performance compared to those created with it.
> > >
> > > Regarding planning functions for SMAC, as you noted, the complexity of the action space makes it challenging for current LLMs to directly implement strategies into executable code. To address this, we simplified the problem by having the LLM decide only whether an agent should move or attack (without specifying which enemy). This approach provides general guidance on actions while circumventing the limitations of directly designing SMAC-compatible outputs.
> > > The reward functions were generated alongside the planning functions using the same prompts. Our approach aimed to explore the potential of LLMs in reward shaping, recognizing that more sophisticated techniques exist in the literature. We provided extensive environmental information in the prompts, enabling the LLM to identify key components for rewards or penalties. Details of the generated reward functions can be found in Appendix E.
> > >
> > > However, we observed that the LLM sometimes struggled to fully grasp the significance of each reward component or appropriately weight them in the final calculation. For example, when prompted, the LLM might generate a reward function in the form r=ax+by+cz, but the coefficients a,b, and c often varied significantly between generations, leading to inconsistent performance. Additionally, without specific examples (few-shot prompting), the LLM occasionally misinterpreted or misused variables.
> > >
> > > In the case of SMAC, the generated reward functions tended to resemble those in the source code, incorporating more detailed rewards rather than sparse outcomes like "team win or lose." However, without tailored prompts or additional examples, the LLM sometimes failed to produce reward functions that consistently aligned with the task objectives.
> > >
> > > [1]G. Papoudakis and L. Schäfer, “Benchmarking Multi-Agent Deep Reinforcement Learning Algorithms in Cooperative Tasks”.
> > >
> > > [2]Y. J. Ma et al., “EUREKA: HUMAN-LEVEL REWARD DESIGN VIA CODING LARGE LANGUAGE MODELS.”  arXiv preprint arXiv:2310.12931, 2024.

---

### Official Review · Reviewer_q81i · 2024-11-04

**Soundness:** 3
**Presentation:** 3
**Contribution:** 2
**Rating:** 3
**Confidence:** 5

**Summary:**

The paper introduces "You Only LLM Once for MARL" (YOLO-MARL), a novel framework that integrates large language models (LLMs) with multi-agent reinforcement learning (MARL) to enhance policy learning in cooperative game environments. The core innovation of YOLO-MARL lies in its one-time interaction with LLMs per game environment, which significantly reduces computational costs and avoids the inefficiencies associated with frequent LLM invocations. The framework utilizes LLMs to generate strategy plans, interpret states, and create planning functions that guide the MARL agents. The approach is tested across various game environments, showing improved performance over traditional MARL algorithms.

**Strengths:**

(1) The use of LLMs in MARL to reduce the frequency of model calls is relatively novel. The one-time interaction strategy for each environment, minimizing computational overhead while leveraging LLM's capability, is an innovative approach.
Addressing the computational inefficiency in MARL through LLMs is of high relevance, especially given the increasing complexity and scale of multi-agent systems. The potential application across different domains enhances the paper's impact.

(2) The experimental setup is robust, employing several environments to validate the effectiveness of YOLO-MARL against established MARL benchmarks.

(3) The paper is well-organized, with clear descriptions of the methodology, experiments, and results. The inclusion of figures, tables, and appendices aids in understanding the proposed framework.

**Weaknesses:**

(1) **The necessity of using MARL settings as opposed to single-agent RL isn't convincingly justified. The claim of innovation specifically in the MARL domain seems overstretched without substantial differentiation from potential single-agent applications.**

(2) The paper fails to discuss several relevant studies that also integrate LLMs with multi-agent systems, which could question the novelty and depth of the literature review. Notably, it omits significant recent works, which could provide crucial context and comparison.

For example:

[1] Hu, Sihao, et al. "A survey on large language model-based game agents." arXiv preprint arXiv:2404.02039 (2024).
multi-agent Cooperation and Communication games:

[2] Gong, Ran, et al. "Mindagent: Emergent gaming interaction." arXiv preprint arXiv:2309.09971 (2023).

[3] Wu, Shuang, et al. "Enhance reasoning for large language models in the game werewolf." arXiv preprint arXiv:2402.02330 (2024).

[4] Zhang, Hongxin, et al. "Building cooperative embodied agents modularly with large language models." arXiv preprint arXiv:2307.02485 (2023).

(3) While the experiments demonstrate faster convergence with YOLO-MARL, the longevity and stability of training are not addressed. More extensive training or delayed results could provide deeper insights into the effectiveness and robustness of the proposed method.

(4) The reliance on human expert and LLM-generated planning functions could limit the applicability in strategic or domain-specific scenarios. The adaptability and accuracy of these functions in complex environments like Go or Dota2 are not discussed.

**Questions:**

(1) Can the authors clarify why MARL is specifically required for the YOLO-MARL framework? Would similar benefits not be realized in single-agent settings?

(2) It is recommended to include and discuss additional related works. How does YOLO-MARL compare with these approaches, especially concerning reward assignment and policy generation in multi-agent contexts?

(3) Could the authors extend their experiments to show long-term training results? It seems that the YOLO-MARL help faster convergence rather than higher performance, especifically in SMAC. Could the authors show longer training results in MPE and LBF environments?

(4) Additionally, testing in environments with higher strategic depth might validate the framework's utility in more complex scenarios.
How does the dependence on pre-trained LLMs affect the generalizability and reliability of the planning functions, especially in domain-specific environments, e.g., chess? The current experimental environments rely on the general pre-training of the LLM to generate a reasonably effective strategy, which while facilitating MARL convergence, might limit the method's upper potential.

(5) **In my view, using human experience-based "State Interpretation" to prompt the LLM, which then generates a base strategy "Planning Function" for RL, seems to be a straightforward application of LLMs in MARL.** This approach does leverage human-like reasoning for initial strategy formulation but might not fully exploit the potential complexities and capabilities of LLMs in dynamic or highly strategic game settings.

---

> ### Author Response · Authors · 2024-11-23
>
> For question about single agent:
> So our study case is mainly focused on multi-agent fully cooperative environment, which I think can’t be realized in single agent settings. The strategies given by LLMs and the generated planning function are mainly focused on the coordination between multi-agents, and the coordination is also the main idea of this paper.
> However, I assume our pipeline can be actually implemented in single agent case. But the extra experiment we provided will also focus on multi-agent setting, since this is the topic of our method. For LLM and single agent RL, as summarized in related work, “Reward Design with Language” designs methods that provide scalar rewards based on suggestions from LLMs to guide RL training. (add single agent RL LLM related work here, that we have summarized in our related work; of course, if there is something new we should cite, just revise the related work section). In contrast, as we motivated our work in the introduction and related work sections, multi-agent RL has remaining challenges such as collaboration among agents, and very limited study on whether LLM is helpful or now in different MARL scenarios. Our work would like to explore this direction, starting from what can be a promising architecture and what limitations exist.
> And we also revised related work in revised version of our paper.

---

### Official Review · Reviewer_jpnq · 2024-11-12

**Soundness:** 1
**Presentation:** 2
**Contribution:** 1
**Rating:** 3
**Confidence:** 4

**Summary:**

This paper proposes a framework that utilizes Large Language Models (LLMs) to facilitate MARL collaboration. By leveraging LLMs for task planning and assigning different tasks to underlying agents, the framework influences policy learning through additional rewards, aiming to accelerate MARL algorithm learning.

**Strengths:**

I believe that using LLMs to enhance collaboration in MARL is a highly promising direction.

**Weaknesses:**

- The methodology throughout the paper doesn’t feel novel to me; I’ve seen several similar studies, and the quality of this paper doesn’t seem satisfactory.
- None of the result figures appear to be processed; it looks as though they were directly downloaded from WANDB. Some figures show results from only one run, lacking statistical significance. I believe at least five runs should be conducted for each result.
- Certain detailed introductions seem unnecessary, such as STATE INTERPRETATION, while others, like PLANNING FUNCTION GENERATION, are not sufficiently elaborated.
- The improvements in experimental results are somewhat expected. However, for slightly more challenging tasks, such as those in SMAC (although the tasks provided are relatively easy), there is no significant performance improvement observed.

**Questions:**

- Could you clarify if the inquiry occurs only once at the beginning of the game?
- Could you provide experimental results on more challenging tasks?

---

> ### Author Response · Authors · 2024-11-23
>
> As you can refer to the Figure 1 in page 2, the interaction with the LLM only occurs at the first time you use a specific new environment. We put this in both abstract and introduction: “Notably, for each game environment, YOLO-MARL only requires one time interaction with LLMs in the proposed strategy generation, state interpretation and planning function generation modules, before the MARL policy training process.” To be more specific, let’s say you want to use our pipeline for a brand new environment E, you will only need to follow the pipeline to prompt LLM at the first time to get generated planning function. After this, you can directly use this planning function in your later MARL training, and you don’t even need to interact with the LLM at the beginning of the game when you start a new episode. We will also submit a revised version of this paper to make the parts you mentioned more clear later.

---

### Author Response · Authors · 2024-11-26
**About SMAC performance**

Again, thank you for your insightful feedback on our manuscript. We appreciate the opportunity to clarify and expand upon our results in the SMAC environment using the YOLO-MARL method.
To the best of our knowledge, we are the first to employ Large Language Models (LLMs) to generate Python planning functions for fully cooperative multi-agent reinforcement learning (MARL) tasks. As this is an initial exploration, we did not claim that our method would significantly outperform existing approaches in the SMAC environment. While the strategies generated by LLMs are logically sound, there can be discrepancies between these strategies and the corresponding Python planning functions produced by the LLMs. This gap may notably affect the final performance outcomes.
In contrast, within the MPE and LBF environments, we observed a high degree of alignment between the generated strategies and the Python planning functions. These results highlight the promising potential of YOLO-MARL, especially as the coding capabilities of LLMs continue to advance. We anticipate that improvements in LLM coding proficiency will further enhance the effectiveness of our approach.
Moreover, we conducted evaluations not only on easy maps like "3m" and "2s_vs_1sc" but also on the challenging map "2c_vs_64zg." This comprehensive testing underscores the versatility and applicability of our framework across varying levels of difficulty within SMAC.
We believe that our work might give some insights of how to use LLMs in MARL tasks and holds potential for future research, particularly as LLM technology evolves.
Thank you once again for your thoughtful review.

---

### Meta-Review · Area_Chair_4Roc · 2024-12-21

**Metareview:**

The paper presents a framework that integrates LLMs with MARL through one-time LLM interaction per environment. The approach uses LLMs for strategy generation, state interpretation, and planning function generation to enhance MARL training. Despite interesting ideas and potential, the paper needs substantial improvements in experimental validation, technical details, and addressing scalability concerns before meeting conference standards.

**Additional Comments On Reviewer Discussion:**

The discussion process revealed both strengths and limitations of the work, with authors being responsive to reviewer concerns while maintaining their position on the value of their approach.

---

### Decision · Program_Chairs · 2025-01-22

Reject